# The protein kinase Ire1 has a Hac1-independent essential role in iron uptake and virulence of *Candida albicans*

**Bernardo Ramírez-Zavala[1], Ines Krüger[1], Christine Dunker[2], Ilse D. Jacobsen[2,3], Joachim Morschhäuser**[1]*

**1** Institute for Molecular Infection Biology, University of Würzburg, Würzburg, Germany, **2** Research Group Microbial Immunology, Leibniz Institute for Natural Product Research and Infection Biology, Hans Knoell Institute, Jena, Germany, **3** Institute of Microbiology, Friedrich Schiller University Jena, Jena, Germany

* joachim.morschhaeuser@uni-wuerzburg.de

**Data Availability Statement:** All relevant data are within the manuscript and its Supporting Information files.

## Abstract

Protein kinases play central roles in virtually all signaling pathways that enable organisms to adapt to their environment. Microbial pathogens must cope with severely restricted iron availability in mammalian hosts to invade and establish themselves within infected tissues. To uncover protein kinase signaling pathways that are involved in the adaptation of the pathogenic yeast *Candida albicans* to iron limitation, we generated a comprehensive protein kinase deletion mutant library of a wild-type strain. Screening of this library revealed that the protein kinase Ire1, which has a conserved role in the response of eukaryotic cells to endoplasmic reticulum stress, is essential for growth of *C. albicans* under iron-limiting conditions. Ire1 was not necessary for the activity of the transcription factor Sef1, which regulates the response of the fungus to iron limitation, and Sef1 target genes that are induced by iron depletion were normally upregulated in *ire1Δ* mutants. Instead, Ire1 was required for proper localization of the high-affinity iron permease Ftr1 to the cell membrane. Intriguingly, iron limitation did not cause increased endoplasmic reticulum stress, and the transcription factor Hac1, which is activated by Ire1-mediated removal of the non-canonical intron in the *HAC1* mRNA, was dispensable for Ftr1 localization to the cell membrane and growth under iron-limiting conditions. Nevertheless, expression of a pre-spliced *HAC1* copy in *ire1Δ* mutants restored Ftr1 localization and rescued the growth defects of the mutants. Both *ire1Δ* and *hac1Δ* mutants were avirulent in a mouse model of systemic candidiasis, indicating that an appropriate response to endoplasmic reticulum stress is important for the virulence of *C. albicans*. However, the specific requirement of Ire1 for the functionality of the high-affinity iron permease Ftr1, a well-established virulence factor, even in the absence of endoplasmic reticulum stress uncovers a novel Hac1-independent essential role of Ire1 in iron acquisition and virulence of *C. albicans*.

**Funding:** This study was funded by the German Research Foundation (DFG) through the TRR 124 FungiNet, "Pathogenic fungi and their human host: Networks of Interaction," DFG project number 210879364, projects C2 to JM and C5 to IDJ. Publication of the work was supported by the Open Access Publication Fund of the University of Würzburg. The funders had no role in study design, data collection and analysis, decision to publish, or preparation of the manuscript. DFG URL: https://www.dfg.de.

**Competing interests:** The authors have declared that no competing interests exist.

## Author summary

The yeast *Candida albicans* is normally a harmless member of the microbial flora in the gastrointestinal tract of healthy people, but when host defenses break down, it can invade into the bloodstream and cause life-threatening disseminated infections. The transition to a pathogenic lifestyle includes radically changed environmental conditions to which the fungus must adapt. In this study, we systematically investigated the role of protein kinases, which are part of virtually all signal transduction pathways that regulate cellular responses to environmental changes, in the adaptation of *C. albicans* to the severely iron-restricted conditions encountered in blood and internal organs. We found that the protein kinase Ire1 is essential for the ability of *C. albicans* to grow under iron-depleted conditions, although it is not required for the upregulation of genes that are induced in response to iron limitation. Instead, Ire1 is critical for the transport of a high-affinity iron permease to the cell membrane, and this function does not require the transcription factor Hac1, which is normally activated by Ire1 when unfolded proteins accumulate in the endoplasmic reticulum. Our study uncovered a novel essential role of this protein kinase in iron uptake and virulence of *C. albicans* beyond its conserved function in the response to endoplasmic reticulum stress.

## Introduction

Microorganisms must be able to obtain all nutrients that are required for growth from their surroundings. Iron is a micronutrient that is essential for almost all organisms but not readily accessible in many environments because of its extremely low solubility in the oxidized state, $Fe^{3+}$, at neutral pH. Iron is also withheld from pathogenic microbes in normally sterile body sites in the human host by sequestration in iron-binding proteins and relocation from sites of infection [1]. Highly efficient iron uptake systems are therefore required for a successful establishment and growth of microorganisms in such environments [2].

The yeast *Candida albicans* is part of the normal microflora in the gastrointestinal tract of most healthy people. However, when immune functions are defective and/or the mucosal barrier breaks down, the fungus can get access to the bloodstream and cause life-threatening systemic infections. Current evidence indicates that iron is easily obtainable for commensal growth of *C. albicans* in at least some parts of the gastrointestinal tract, such that its uptake has to be limited to avoid toxic effects of excess iron within the cells [3]. In contrast, virtually no free iron is available in blood and internal organs, and *C. albicans* depends on efficient iron acquisition mechanisms to be able to cause a systemic infection [4]. Similar to other fungi, *C. albicans* possesses a high-affinity iron transporter, Ftr1, that enables growth when iron is severely limited [5]. Ftr1 associates with one of several redundant multicopper ferroxidases to form a functional iron permease. The ferroxidase reoxidizes reduced $Fe^{2+}$, generated by cell membrane-located iron reductases, and directly channels the produced $Fe^{3+}$ to Ftr1 [6–9]. In *C. albicans*, Ftr1 has a key role during a systemic infection, because *ftr1Δ* mutants are avirulent in the mouse model of disseminated candidiasis [5]. In addition, *C. albicans* can utilize heme from hemoglobin as an iron source via a specific transport system, and mutants lacking components of this system also display reduced virulence in a mouse model of systemic candidiasis [10]. *C. albicans* can also extract iron from other host proteins, such as ferritin and transferrin, using specific surface proteins and the reductive iron assimilation pathway, and it can furthermore utilize iron-containing siderophores produced by other microorganisms [11–13].

The genes that are required for iron uptake and utilization in *C. albicans* are controlled by a regulatory circuit composed of three transcription factors [3]. Under iron-replete conditions, the GATA transcription factor Sfu1 represses the expression of iron uptake genes and of *SEF1*, encoding a transcriptional activator of the zinc cluster transcription factor family. When iron becomes limiting, Sef1 activates the expression of iron uptake genes and of *HAP43*, which encodes a repressor that downregulates genes involved in iron-utilizing processes. Hap43 also downregulates *SFU1*, thereby relieving *SEF1* from Sfu1-mediated transcriptional repression. Sfu1 inhibits Sef1 activity also at a posttranscriptional level, by protein complex formation and retention in the cytosol, where Sef1 is destabilized. Under iron-limiting conditions, Sef1 forms an alternative complex with the protein kinase Ssn3, which results in Sef1 phosphorylation, nuclear localization, and transcriptional activation of its target genes [14]. Sef1 is essential for *C. albicans* to adapt to iron limitation, because *sef1Δ* mutants are unable to grow in iron-depleted media, and the mutants are also highly attenuated for virulence in a mouse model of systemic infection [3].

In a previous study from our lab we had used a library of artificially activated forms of all *C. albicans* zinc cluster transcription factors, generated by fusing the Gal4 activation domain (GAD) to their C-terminus, to uncover their biological roles and reveal novel phenotypes conferred by gain-of-function mutations in these transcriptional regulators [15]. Strains expressing the hyperactive *SEF1-GAD* allele produced a yellow culture supernatant when grown in minimal medium due to the secretion of riboflavin, which occurs in *C. albicans* and other yeasts in response to iron limitation [16–18]. To gain more insight into the control of Sef1 activity and the signaling pathways enabling *C. albicans* to adapt to iron limitation, we tested if the hyperactive Sef1 would bypass the reported requirement of the protein kinase Ssn3 for Sef1 function and growth under iron-depleted conditions. As described below, we unexpectedly observed that Ssn3 is dispensable for Sef1 functionality and adaptation to iron limitation. We therefore performed a systematic analysis of protein kinase signaling pathways of *C. albicans*, which revealed the essential role of Ire1 for growth in an iron-limited environment and uncovered its molecular basis.

## Results

### Ssn3 is not required for Sef1-induced riboflavin secretion and growth under iron-limiting conditions

Since the protein kinase Ssn3 has been shown to be required for the functionality of Sef1 [14], we tested if Ssn3 is also needed for riboflavin secretion in response to iron limitation and if a hyperactive form of Sef1 overcomes its dependence on Ssn3. To this aim, the *SEF1-GAD* allele (henceforth designated *SEF1\**) was integrated into the genome of two independently generated *ssn3Δ* mutants of the wild-type strain SC5314 and expressed from the *ADH1* promoter, in the same way as in the parental strain SC5314 [15]. For comparison, wild-type *SEF1* was also overexpressed from the *ADH1* promoter in both strain backgrounds. As shown in Fig 1A, overexpression of *SEF1\** or wild-type *SEF1* promoted riboflavin secretion under iron-replete conditions, and this occurred independently of the presence of Ssn3 (the constitutive overexpression of *SEF1* or *SEF1\** also had a slight negative effect on the growth of the strains, Fig 1B). Unexpectedly, Ssn3 was also not required for growth and riboflavin secretion under iron-limiting conditions (Fig 1A and 1B). We confirmed that Sef1 was essential for growth in the iron-depleted medium used in our experiments, by testing two independently generated *sef1Δ* mutants of strain SC5314. Both homozygous *sef1Δ* mutants were unable to grow in the iron-depleted medium, and growth was restored after reintegration of a functional *SEF1* allele into its endogenous genomic locus (Fig 2A and 2B). Riboflavin secretion was already impaired in

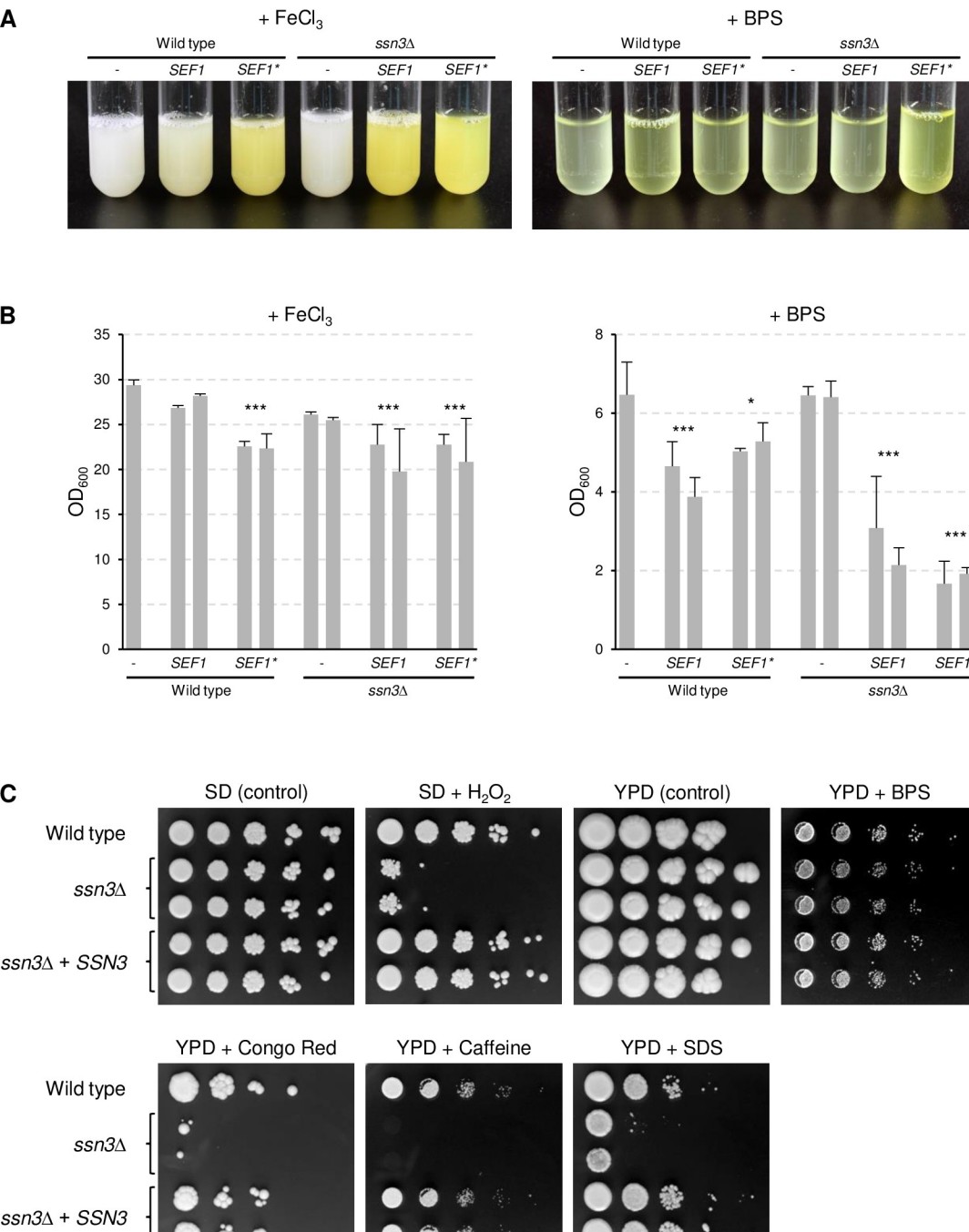

**Fig 1. Ssn3 is not required for Sef1-induced riboflavin secretion and growth under iron-limiting conditions.** (A) The wild-type strain SC5314, *ssn3Δ* mutants, and derivatives overexpressing wild-type *SEF1* or the hyperactive *SEF1\** were grown in minimal medium with (+ FeCl₃) or without iron (+ BPS). Cultures were photographed after 4 days of growth at 30°C. The two independently generated series of strains behaved identically and only one of them is shown. Removal of the cells by centrifugation (not shown) demonstrated that the yellow riboflavin was secreted into the supernatant. (B) The optical densities of the cultures were determined; shown are the means and standard deviations of three independent cultures of each strain. Results for both independently generated series of strains are shown separately; data were combined for statistical analysis. Significant differences from the wild type are indicated by stars (***, p < 0.001; *, p < 0.05). (C) Stress sensitivity of the wild-type strain SC5314, *ssn3Δ* mutants, and complemented strains. YPD overnight cultures of the strains were serially 10-fold diluted, spotted on the indicated agar plates, and grown for 4 days at 30°C. Both series of mutants are shown.

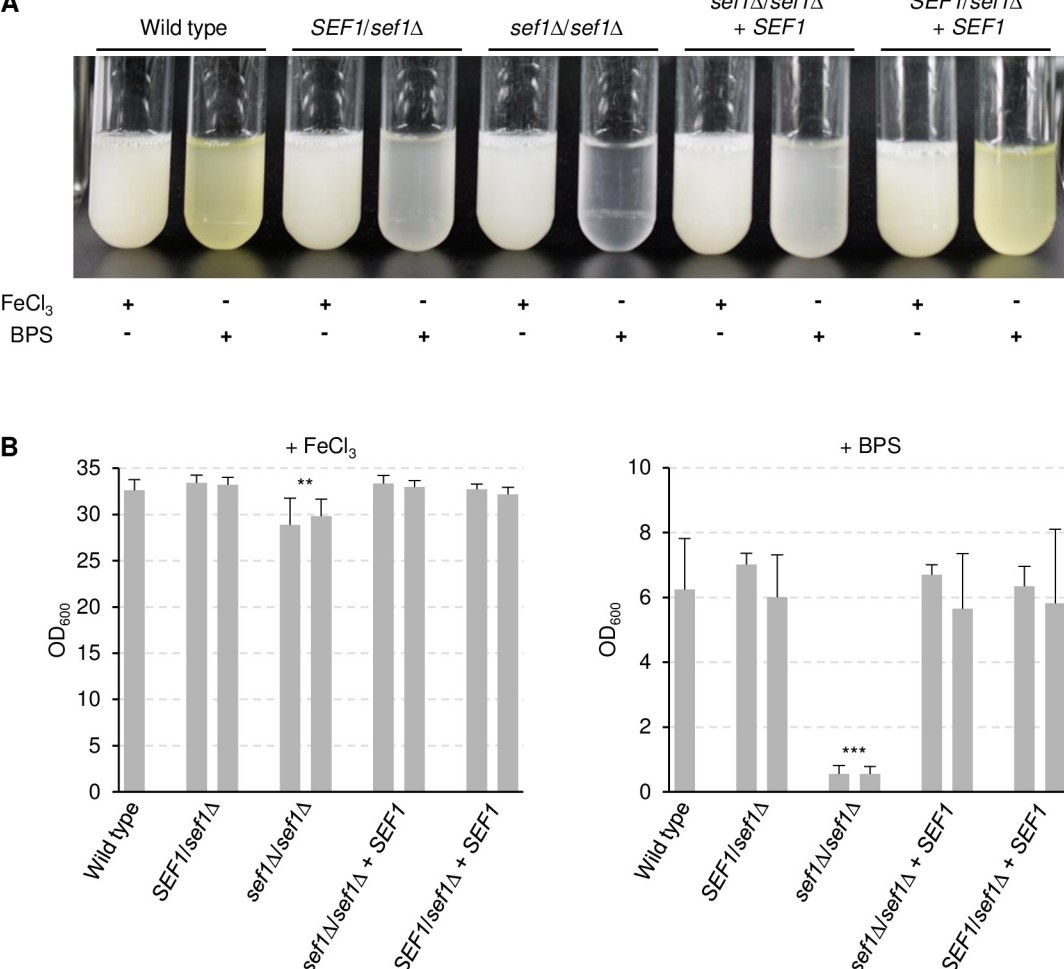

**Fig 2. Sef1 is required for growth and riboflavin secretion in iron-depleted medium.** (A) The wild-type strain SC5314, heterozygous and homozygous *sef1Δ* mutants, and complemented strains in which a wild-type *SEF1* copy was reinserted into the homozygous (*sef1Δ/sef1Δ + SEF1*) or heterozygous (*SEF1/sef1Δ + SEF1*) mutants were grown in minimal medium with (+ FeCl₃) or without iron (+ BPS). Cultures were photographed after 4 days of growth at 30°C. The two independently generated series of strains behaved identically and only one of them is shown. (B) The optical densities of the cultures were determined; shown are the means and standard deviations of three independent cultures of each strain. Results for both independently generated series of strains are shown separately; data were combined for statistical analysis. Significant differences from the wild type are indicated by stars (\*\*\*, p < 0.001; \*\*, p < 0.01).

the heterozygous *SEF1/sef1Δ* mutants and restored when the deleted allele was reinserted, demonstrating a *SEF1* gene dosage effect on iron limitation-induced riboflavin secretion (Fig 2A). Because of the unanticipated dispensability of Ssn3 for growth in iron-depleted medium, we verified that our *ssn3Δ* mutants exhibited other phenotypes described in the literature. Fig 1C shows that the *ssn3Δ* mutants displayed the previously reported hypersensitivity of an *ssn3* transposon insertion mutant to hydrogen peroxide [19], and a slightly reduced growth was also observed on YPD plates containing 300 mM of the iron chelator bathophenanthroline disulfonate (BPS), confirming that Ssn3 is required for normal growth on iron-depleted rich solid medium [14]. The *ssn3Δ* mutants were also hypersensitive to cell wall/membrane stress-inducing agents (Congo Red, caffeine, SDS), and all growth defects were complemented after reintroduction of a functional *SSN3* copy into the homozygous mutants (Fig 1C).

## Generation of a comprehensive *C. albicans* protein kinase deletion mutant library

Sef1 is phosphorylated at several serine and threonine residues [20], and a previous study has shown that Sef1 phosphorylation occurs upon iron depletion in an Ssn3-dependent manner [14]. Since Ssn3 was dispensable for Sef1 function in our experiments, we wanted to investigate if other protein kinases might be involved in the regulation of Sef1 activity. The *C. albicans* genome contains 108 genes encoding known or predicted protein kinases, and in a previous study we had constructed a limited set of deletion mutants of the prototrophic wild-type reference strain SC5314 lacking uncharacterized protein kinases [21]. To enable a systematic search for protein kinases that are required for Sef1 function and adaptation of *C. albicans* to iron limitation (as well as other future systematic investigations of protein kinase function in *C. albicans*), we completed this library in our present study. The mutants were generated using the *SAT1*-flipping strategy, which is based on the sequential replacement of both alleles of the target gene by the dominant *caSAT1* selection marker and subsequent excision of the marker from the genome by the site-specific recombinase FLP [21,22]. This approach avoids potential unspecific effects caused by the use of auxotrophic markers and host strains, and the final mutants are truly isogenic and can be directly compared with the parental wild-type strain. For each target gene, two independent series of deletion mutants were generated to ensure the reproducibility of phenotypes caused by the absence of a protein kinase (see S1 Table). For 81 of the 108 genes, the two alleles in strain SC5314 were distinguished by a restriction fragment length polymorphism and both possible heterozygous mutants were retained and used for the construction of the homozygous mutants. We successfully generated homozygous deletion mutants for 86 of the 108 protein kinases. For the remaining 22 protein kinases, only heterozygous mutants were obtained, either because the genes are essential or because of technical difficulties.

## The protein kinases Ire1, Mec1, and Vps15 are required for the adaptation of *C. albicans* to iron limitation

We tested the ability of the protein kinase deletion mutants to grow and secrete riboflavin in iron-depleted medium. In a first qualitative screening of the library, we identified three mutants (*ire1Δ*, *mec1Δ*, *vps15Δ*) that had a severe growth defect in minimal medium containing the iron chelator BPS and did not detectably secrete riboflavin. These mutants were selected for a detailed investigation. Several other mutants had severe growth defects also in the absence of BPS or even in rich YPD medium and were not further studied.

To confirm that the deletion of *IRE1*, *MEC1*, and *VPS15* caused a growth defect in iron-depleted media, we reintroduced a wild-type copy of the genes into the respective mutants and tested growth and riboflavin secretion of each series of strains. As can be seen in Fig 3A and 3B, all homozygous mutants exhibited a severe growth defect in iron-depleted minimal medium and, consequently, did not produce detectable levels of riboflavin. Growth and riboflavin secretion were restored upon reintroduction of a wild-type gene copy. Of note, while the *ire1Δ* and *mec1Δ* mutants showed nearly wild-type growth in iron-replete medium, the *vps15Δ* mutants displayed a more strongly reduced growth also in the control medium. Similar growth phenotypes were observed on YPD plates in the absence and presence of BPS (Fig 3C). The *ire1Δ* and *vps15Δ* mutants were unable to grow on the iron-depleted medium, whereas the growth defect of the *mec1Δ* mutants was milder. On the control plates, the *ire1Δ* and *mec1Δ* mutants grew well, but the latter displayed a slightly filamentous colony phenotype. The *vps15Δ* mutants showed reduced growth also on rich solid medium.

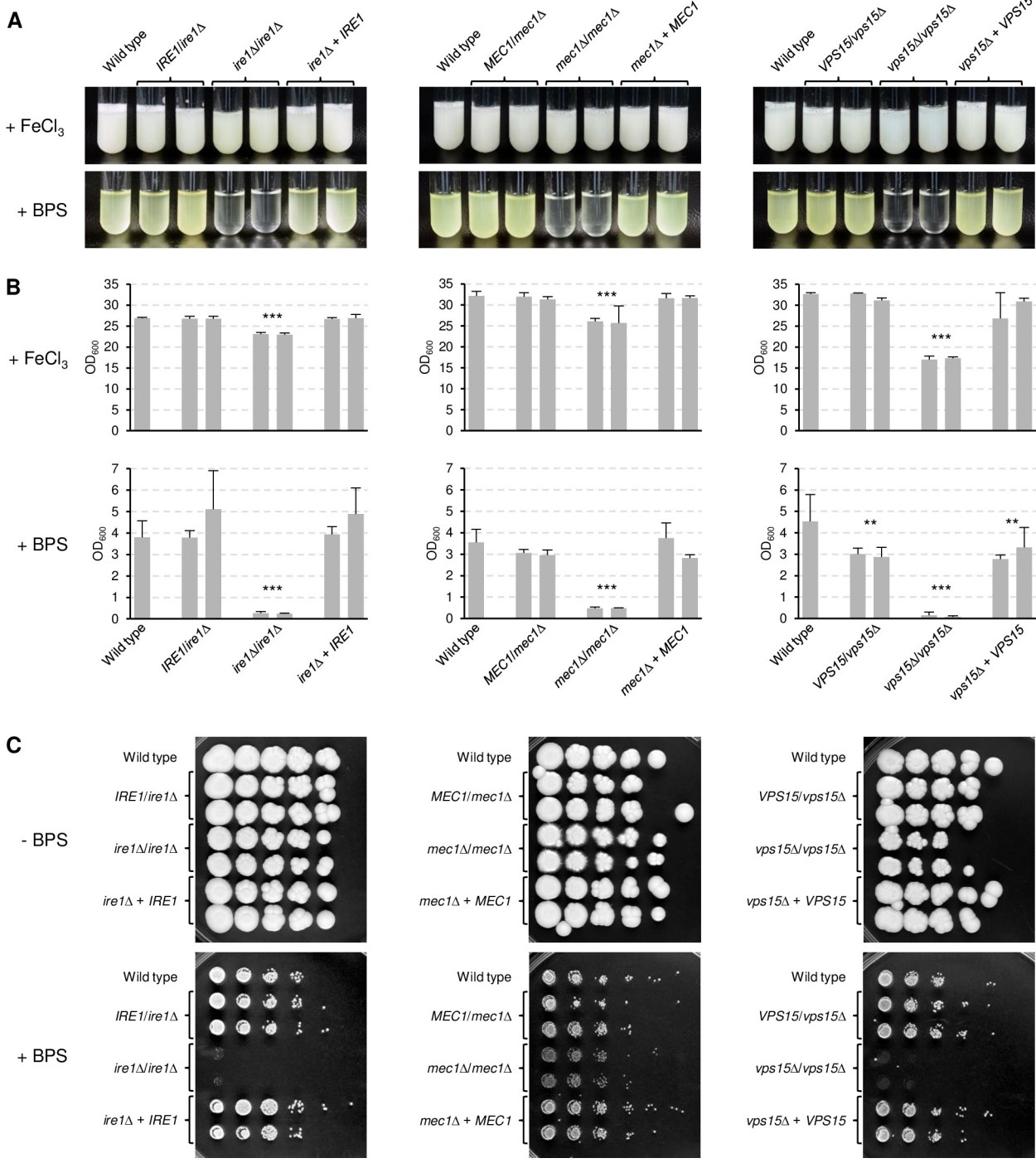

**Fig 3. The protein kinases Ire1, Mec1, and Vps15 are required for growth under iron-limiting conditions.** (A) The wild-type strain SC5314, heterozygous and homozygous *ire1Δ*, *mec1Δ*, and *vps15Δ* mutants, and complemented strains were grown in minimal medium with (+ FeCl₃) or without iron (+ BPS). Cultures were photographed after 4 days of growth at 30˚C. Both independently generated series of strains are shown. (B) The optical densities of the cultures were determined; shown are the means and standard deviations of three independent cultures of each strain. Data for both independently generated series of strains were combined for statistical analysis. Significant differences from the wild type are indicated by stars (***, p < 0.001). (C) YPD overnight cultures of the strains were serially 10-fold diluted, spotted on YPD agar plates without or with BPS, and grown for 4 days at 30˚C.

## Ire1, Mec1, and Vps15 are dispensable for Sef1 functionality

We reasoned that, if the inability of the *ire1Δ*, *mec1Δ*, and *vps15Δ* mutants to adapt to iron limitation was caused by defective Sef1 activation, overexpression of wild-type *SEF1* or the hyperactive *SEF1** might bypass the requirement of these kinases for growth under iron limitation. Fig 4 shows that *SEF1* or *SEF1** overexpression in the mutants resulted in constitutive riboflavin secretion under iron-replete conditions, similar to its effect in the wild type. Therefore, at least when overexpressed, Sef1 was functional in the absence of Ire1, Mec1, or Vps15. However, overexpression of *SEF1* or *SEF1** did not restore growth of the *ire1Δ* and *vps15Δ* mutants in iron-depleted medium, indicating that their growth defect was not caused because the kinases were required for Sef1 functionality. In contrast, growth and concomitant riboflavin secretion in iron-depleted medium were improved by *SEF1* or *SEF1** overexpression in the *mec1Δ* mutants, demonstrating that the defect of these mutants can at least partially be bypassed by increased Sef1 activity.

We then examined if the induction of known Sef1 target genes [3] in response to iron depletion depends on Ire1, Mec1, or Vps15. *FTR1* encodes a high-affinity iron permease that is required for growth in iron-deficient medium [5], *FRP1* a ferric reductase that is strongly upregulated in low-iron conditions [23], *FET31* a multicopper oxidase that is required for growth under low-iron conditions [24], *CSA1* a cell surface protein that is strongly induced under iron-limiting conditions [3], and *RIB1* the rate-limiting enzyme of riboflavin biosynthesis [17]. Northern hybridization experiments (Fig 5A) demonstrated that the upregulation of all five tested genes upon iron depletion did not require any of the three protein kinases, confirming that Sef1 remains functional in their absence.

Finally, we investigated if the Sef1 protein is affected in the absence of Ire1, Mec1, or Vps15. To detect Sef1, we added a 3xHA-tag to one of the endogenous *SEF1* alleles in the wild type and the kinase mutants. Sef1-HA protein levels were increased in response to iron depletion in the wild type as well as in the kinase mutants, although the increase was weaker in the *vps15Δ* mutants compared to the other strains (Fig 5B). A slight shift to a lower-mobility form of the HA-tagged Sef1 was observed in the wild-type strain, in line with the previously reported phosphorylation of Sef1 upon iron depletion [14]. This shift was also observed in the *ire1Δ* and *mec1Δ* mutants, but less clear in the *vps15Δ* mutants. Altogether, these results indicated that Ire1, Mec1, and Vps15 do not act as upstream activating kinases for Sef1, and the inability of mutants lacking these kinases to adapt to iron-limiting conditions is caused by other defects.

## Ire1 is required for cell membrane localization of the high-affinity iron permease Ftr1

Since Sef1 function and the transcriptional response to iron limitation were not affected in the *ire1Δ*, *mec1Δ*, and *vps15Δ* mutants, we investigated if the absence of the kinases directly impacted proteins required for iron utilization, such as the high-affinity iron permease Ftr1. We fused one of the endogenous *FTR1* alleles with *GFP* in the wild type and protein kinase mutants and observed the subcellular localization of the C-terminally GFP-tagged Ftr1 upon iron starvation by fluorescence microscopy. As can be seen in Fig 6A, Ftr1-GFP was specifically produced during growth in iron-depleted medium and localized at the cell periphery in the wild type, as expected, and also in the *mec1Δ* mutants. Ftr1-GFP was also correctly localized in the *vps15Δ* mutants, although it was produced in lower amounts and surprisingly also in the absence of the iron chelator. Most strikingly, however, Ftr1-GFP was mislocalized in the *ire1Δ* mutants and mainly detected at intracellular sites instead of the plasma membrane. Since Ftr1-GFP levels in the *vps15Δ* mutants, as observed by fluorescence microscopy, did not match the expression pattern of *FTR1* mRNA (see Fig 5A), we compared Ftr1-GFP levels in the

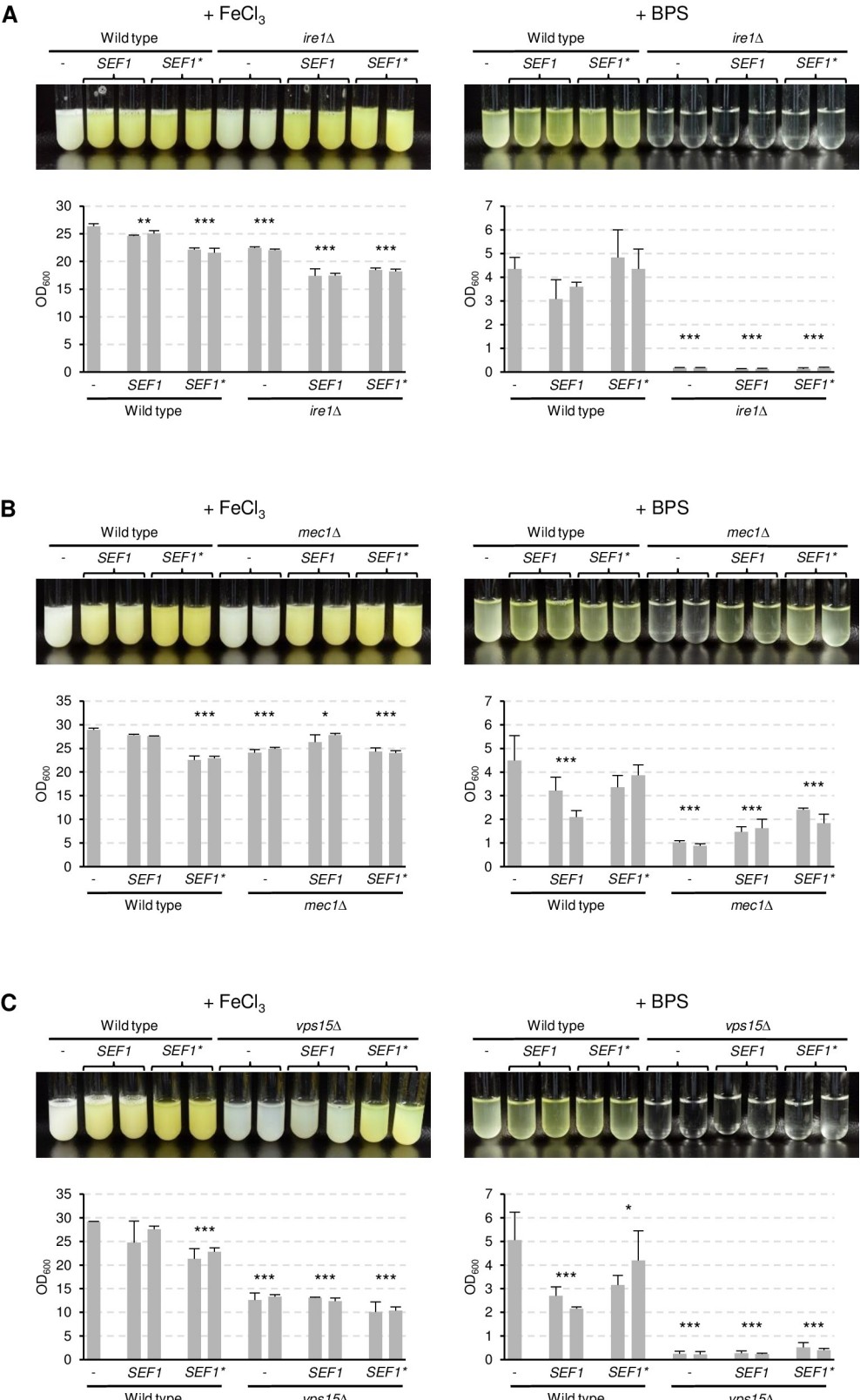

**Fig 4. Sef1 is functional in the absence of Ire1, Mec1, and Vps15.** Strains overexpressing wild-type *SEF1* or the hyperactive *SEF1\** in *ire1Δ* (A), *mec1Δ* (B), and *vps15Δ* (C) backgrounds were grown in minimal medium with

(+ FeCl₃) or without iron (+ BPS); corresponding wild-type controls were included for comparison in each
experiment. Cultures were photographed after 4 days of growth at 30˚C (top panels) and the optical densities of three
independent cultures of each strain determined (bottom panels showing means and standard deviations). Results for
both independently generated series of strains are shown separately; data were combined for statistical analysis.
Significant differences from the wild type are indicated by stars (***, p < 0.001; **, p < 0.01; *, p < 0.05).

various strains by Western blotting (Fig 6B). Similar to the fluorescence microscopy results,
Ftr1-GFP protein levels were lower in the *vps15Δ* mutants than in the other strains in iron-
depleted medium but also observable in the control medium. Therefore, the *FTR1-GFP* tran-
script may be more efficiently translated or the Ftr1-GFP protein be more stable in the *vps15Δ*
mutants when sufficient iron is available, but not in its absence.

Ire1 has a conserved role in the unfolded protein response (UPR), which is activated upon
accumulation of unfolded proteins in the endoplasmic reticulum (ER) in organisms from
yeast to mammals [25]. We therefore speculated that Ftr1 might be retained in the ER in the
*ire1Δ* mutants, explaining its irregular localization pattern. Indeed, ER staining of *ire1Δ* cells
containing Ftr1-GFP showed that the GFP signals overlapped with the ER stain, but not with

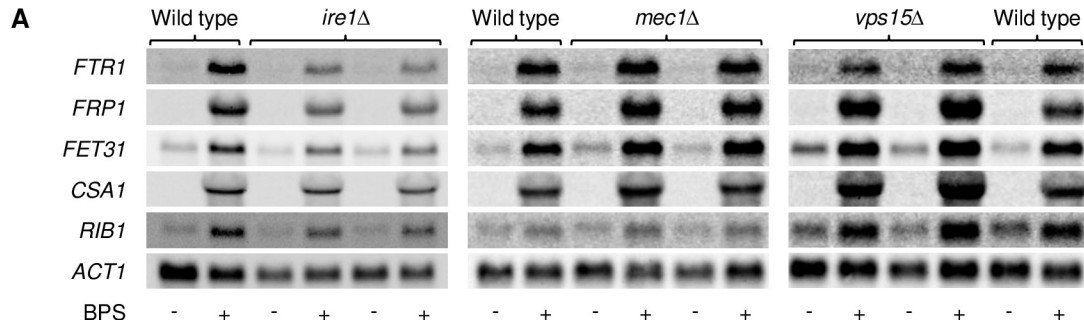

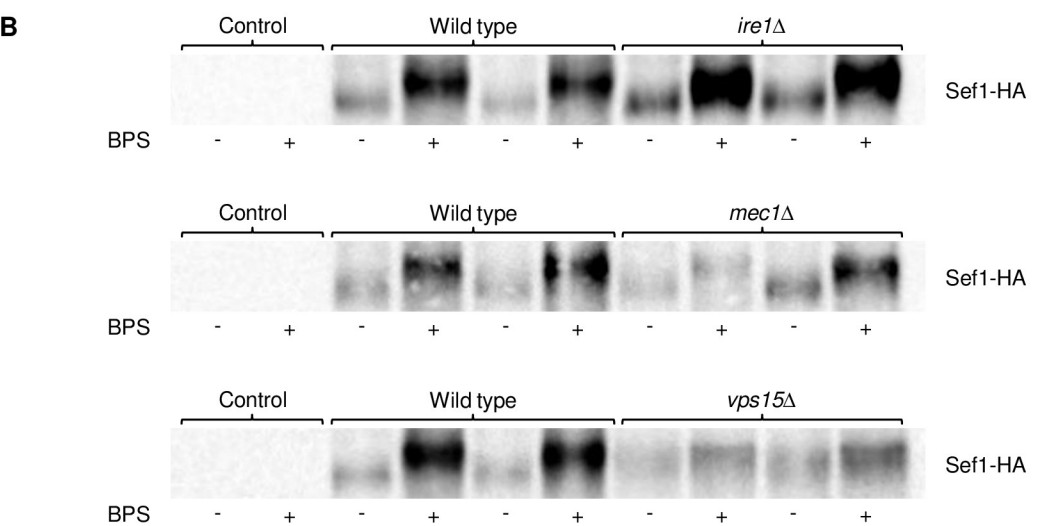

**Fig 5. Upregulation of Sef1 and Sef1 target genes by iron depletion does not depend on Ire1, Mec1, and Vps15.** (A)
Overnight cultures of the wild-type strain SC5314 and *ire1Δ*, *mec1Δ*, and *vps15Δ* mutants were inoculated in fresh YPD
medium without or with BPS and grown for 5 h at 30˚C. Expression of the indicated iron-regulated genes was analyzed by
Northern hybridization with gene-specific probes; *ACT1* mRNA levels served as controls. Each set of experiments was
performed separately. (B) The wild-type strain SC5314 and *ire1Δ*, *mec1Δ*, and *vps15Δ* mutants carrying a 3xHA-tagged *SEF1*
allele were grown as described above and Sef1-HA was detected by Western blotting with an HA-specific antibody. The
untagged wild-type strain served as negative control. Both series of mutants are shown in (A) and (B).

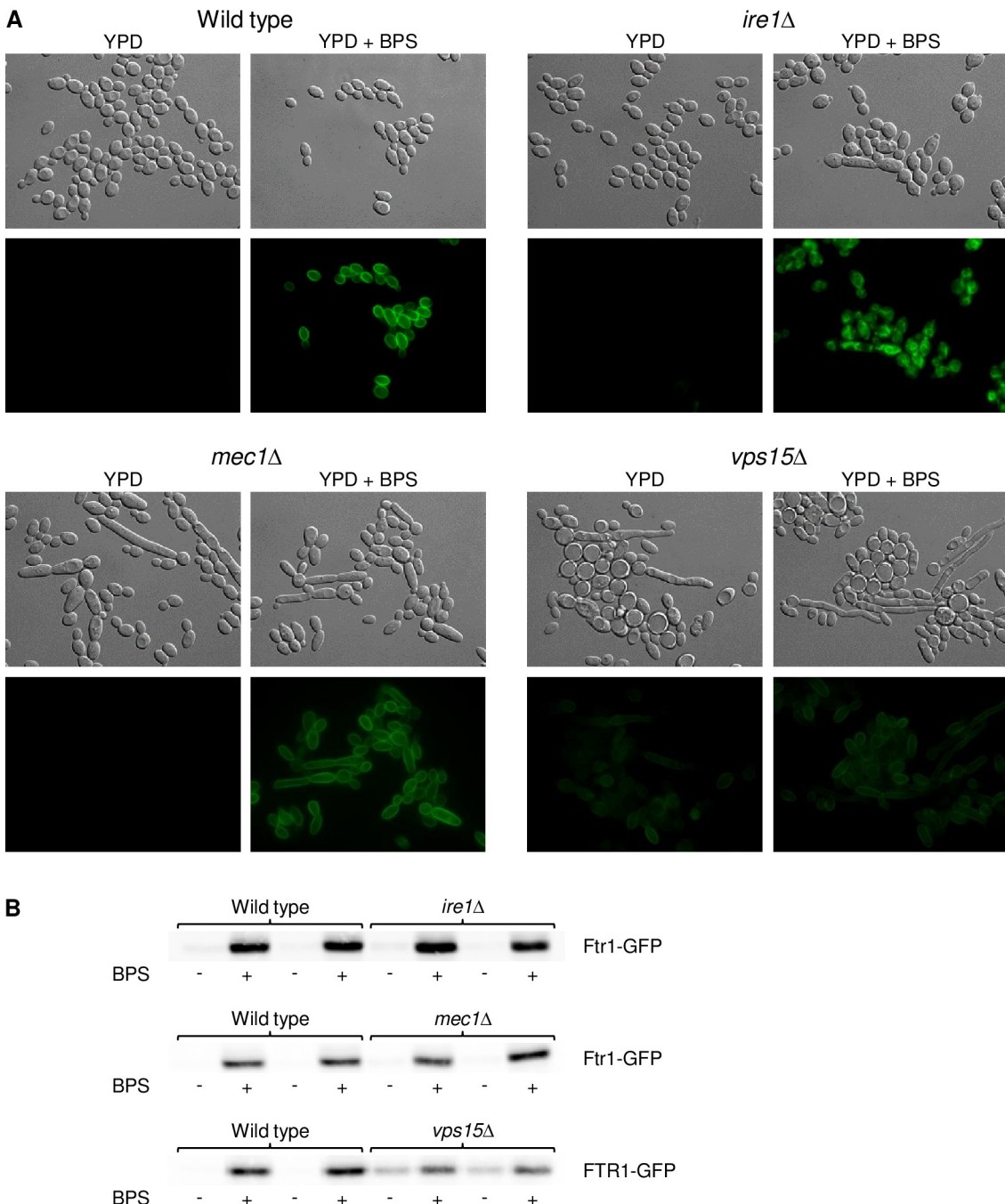

**Fig 6. The high-affinity iron permease Ftr1 is mislocalized in *ire1Δ* mutants.** (A) Overnight cultures of the wild-type strain SC5314 and *ire1Δ*, *mec1Δ*, and *vps15Δ* mutants carrying a *GFP*-tagged *FTR1* allele were inoculated in fresh YPD medium without or with BPS and grown for 5 h at 30°C. Cells were imaged by DIC and fluorescence microscopy. The two independently generated series of strains behaved identically and only one of them is shown. (B) Strains were grown as described above and analyzed by Western blotting with an anti-GFP antibody. Results for both independently generated series of strains are shown in each case.

the vacuole-staining dye FM4-64, supporting this hypothesis (Fig 7). We wondered if Ire1 is generally required for the correct localization of cell membrane-localized nutrient transporters in response to nutrient limitation. We thus investigated if the ammonium permease Mep2, which is strongly upregulated under nitrogen-limiting conditions [26], is also mislocalized in

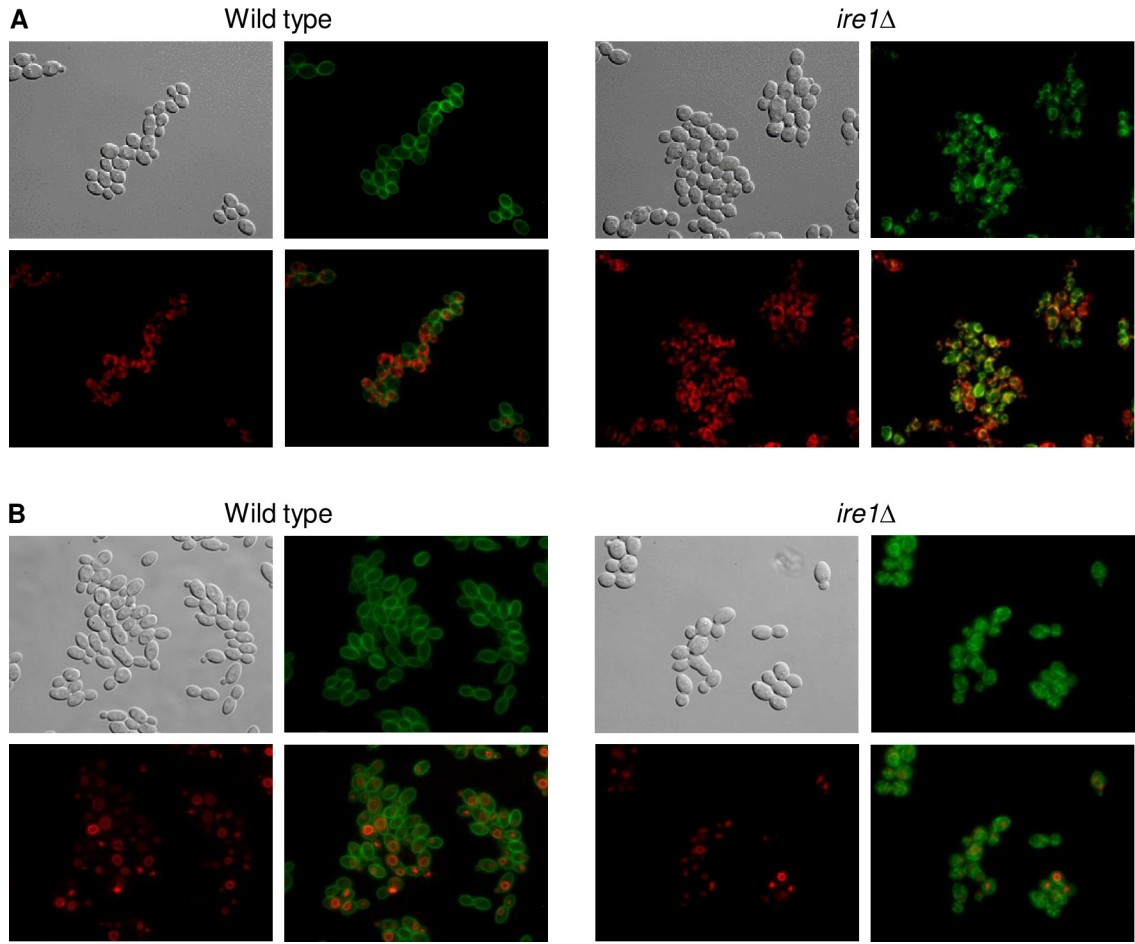

**Fig 7. Ftr1 is retained in the endoplasmic reticulum in *ire1*Δ mutants.** Overnight cultures of the wild-type strain SC5314 and *ire1*Δ mutants carrying a *GFP*-tagged *FTR1* allele were inoculated in fresh YPD medium with BPS, grown for 4 h at 30°C, and stained with ER detection reagent (A) or FM4-64 (B), as described in Materials and methods. Cells were imaged by DIC and fluorescence microscopy using appropriate filters for green and red fluorescence. Shown are DIC images of the cells (top left panels), Ftr1-GFP (top right panels), ER or vacuole staining (bottom left panels), and merged images of Ftr1-GFP and ER/vacuole staining (bottom right panels). The two independently generated series of strains behaved identically and only one of them is shown.

the absence of the kinase. However, GFP-tagged Mep2 was upregulated and correctly localized in the *ire1*Δ mutants, as in the wild type and the other kinase mutants (Fig 8A). We considered the possibility that iron-depletion, but not nitrogen limitation, might cause ER stress and therefore require Ire1 for correct localization of transport proteins in the cell membrane. Therefore, we observed Ftr1-GFP and Mep2-GFP localization under conditions of both iron and nitrogen limitation. Only Ftr1-GFP was mislocalized under these conditions in the *ire1*Δ mutants, while Mep2-GFP showed normal localization (Fig 8B). These results demonstrate that Ire1 is required for cell membrane localization of Ftr1, but not Mep2, and therefore not generally important for correct targeting of appropriate nutrient transporters under conditions of nutrient stress.

## A pre-spliced form of *HAC1* bypasses the requirement of Ire1 for growth under iron-depleted conditions

Since the *mec1*Δ mutants had a milder growth defect than the *ire1*Δ mutants in iron-depleted medium, and the *vps15*Δ mutants also exhibited considerable growth defects in control media

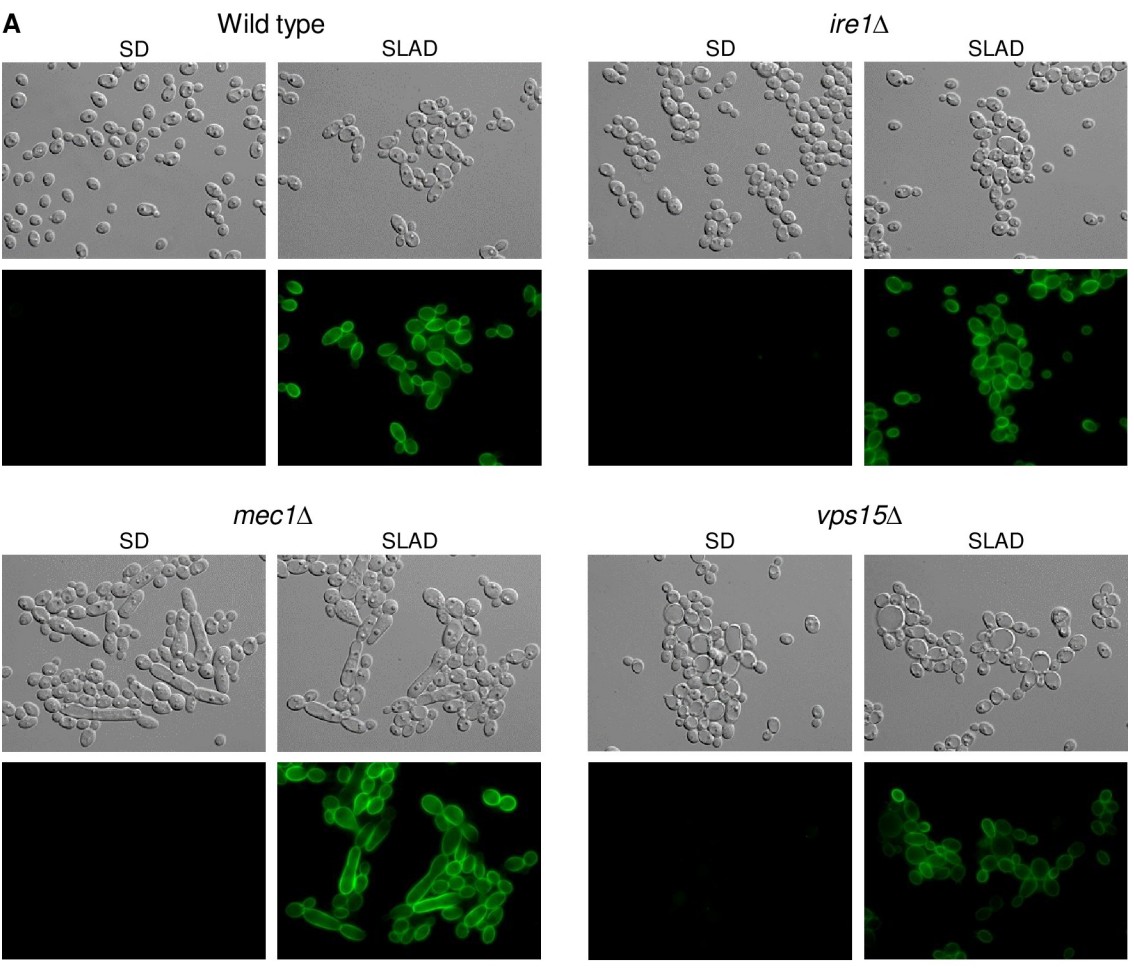

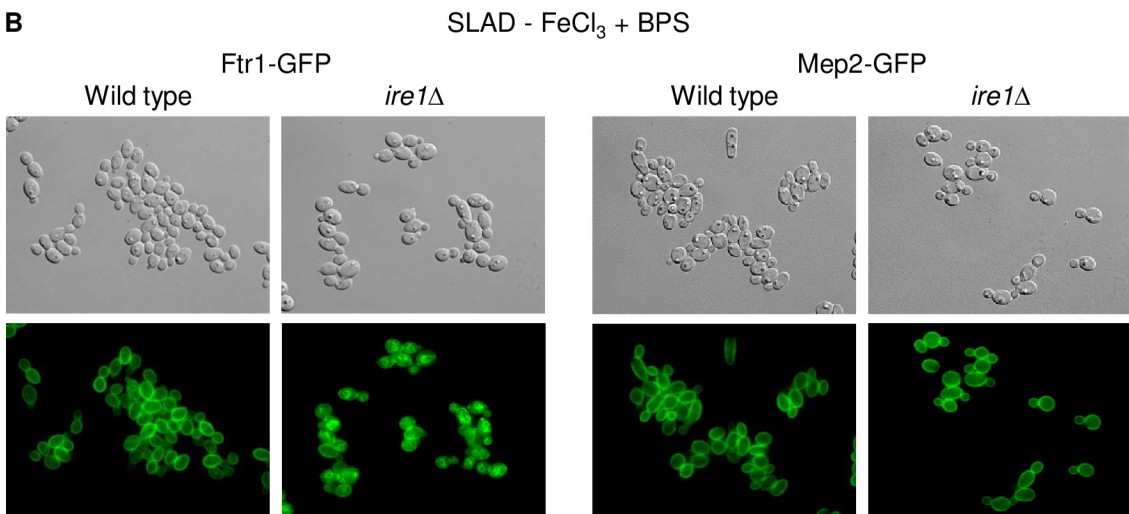

**Fig 8. Ire1 is not required for correct localization of the ammonium permease Mep2 in response to nitrogen limitation.** (A) Overnight cultures of the wild-type strain SC5314 and *ire1Δ*, *mec1Δ*, and *vps15Δ* mutants carrying a *GFP*-tagged *MEP2* allele were inoculated in SD or SLAD medium and grown for 5 h at 30˚C. Cells were imaged by DIC and fluorescence microscopy. The two independently generated series of strains behaved identically and only one of them is shown. (B) The wild type and *ire1Δ* mutants with *GFP*-tagged *FTR1* or *MEP2* alleles were grown in SLAD medium without iron and the cells observed as described above.

(see Figs 3 and 4), we focused our further investigations on the role of Ire1 in the adaptation of *C. albicans* to iron limitation. Ire1 also has an endonuclease activity, and in response to ER stress removes an unusual intron from the *HAC1* mRNA, which allows the production of the active form of the encoded transcription factor and expression of its target genes [27]. In *C. albicans*, the *HAC1* intron is only 19 nucleotides long and excised upon ER stress [28]. If the inability of the *ire1Δ* mutants to adapt to iron limitation was caused by defective *HAC1* mRNA processing, expression of a pre-spliced *HAC1* should bypass their growth defect in iron-depleted medium. To test this possibility, we introduced a mutated, intronless *HAC1* copy (for simplicity referred to as *HAC1\**) under control of the *ADH1* promoter into the *ire1Δ* mutants. Expression of the pre-spliced *HAC1\** did not significantly impact growth of the wild-type parental strain, but restored growth and riboflavin secretion in iron-depleted medium in the *ire1Δ* mutants (Fig 9A). Expression of an additional copy of wild-type *HAC1* from the *ADH1* promoter did not rescue the mutant phenotype, indicating that the inability of the *ire1Δ* mutants to adapt to iron limitation was caused by a defect in *HAC1* mRNA processing. We then tested if the pre-spliced *HAC1\** would enable the cells to correctly target Ftr1 to the cell membrane. This was indeed the case (Fig 9B), suggesting that mislocalization of the high-affinity iron permease Ftr1 (and possibly other proteins required for iron utilization) prevented growth of *C. albicans* cells lacking the Ire1 kinase under iron-limiting conditions.

### *HAC1* is not required for growth of *C. albicans* under iron-limiting conditions

The ability of a pre-spliced *HAC1\** to restore growth of *ire1Δ* mutants in iron-depleted medium suggested that iron limitation results in ER stress that requires Ire1-dependent *HAC1* mRNA splicing. We therefore used reverse transcription PCR (RT-PCR) to detect unspliced and spliced *HAC1* transcripts in the wild type and *ire1Δ* mutants grown in iron-replete and iron-depleted media. As a positive control, the strains were also grown in the presence of DTT and tunicamycin, which are commonly used to induce ER stress. As can be seen in Fig 10A, partial *HAC1* mRNA splicing occurred already under unstressed conditions in wild-type cells, in line with a previous report [28]. Treatment with DTT or tunicamycin converted most *HAC1* transcripts into the spliced form, but no increased splicing occurred in response to iron depletion. As expected, no *HAC1* mRNA splicing was observed in the *ire1Δ* mutants under any condition, confirming that Ire1 mediates *HAC1* mRNA splicing also in *C. albicans*. These results indicated that iron limitation does not cause enhanced ER stress and that basal Ire1-dependent *HAC1* mRNA splicing is required for proper Ftr1 localization and growth of *C. albicans* under iron-limiting conditions.

If the inability of *ire1Δ* mutants to produce a functional Hac1 transcription factor was responsible for their growth defect in iron-depleted media, cells lacking Hac1 should exhibit the same phenotype. We therefore deleted *HAC1* in the wild-type strain SC5314 and compared growth and riboflavin production under iron-limiting conditions in *ire1Δ* and *hac1Δ* mutants. Surprisingly, unlike the *ire1Δ* mutants, the *hac1Δ* mutants displayed wild-type growth and riboflavin production in iron-depleted medium (Fig 10B). Furthermore, a GFP-tagged Ftr1 showed normal induction and subcellular localization in response to iron limitation in the *hac1Δ* mutants (Fig 10C). These results demonstrate that Ire1 can mediate proper localization of Ftr1 and growth under iron-limiting conditions in the absence of Hac1, but a constitutively active Hac1 can bypass the requirement for Ire1 to adapt to iron starvation (see Fig 9).

We then compared the phenotypes of *ire1Δ* and *hac1Δ* mutants under conditions that require an appropriate response to ER and cell wall/membrane stress. Fig 10D shows that, similar to *ire1Δ* mutants, the *hac1Δ* mutants were hypersensitive to DTT, SDS, and Congo Red, in

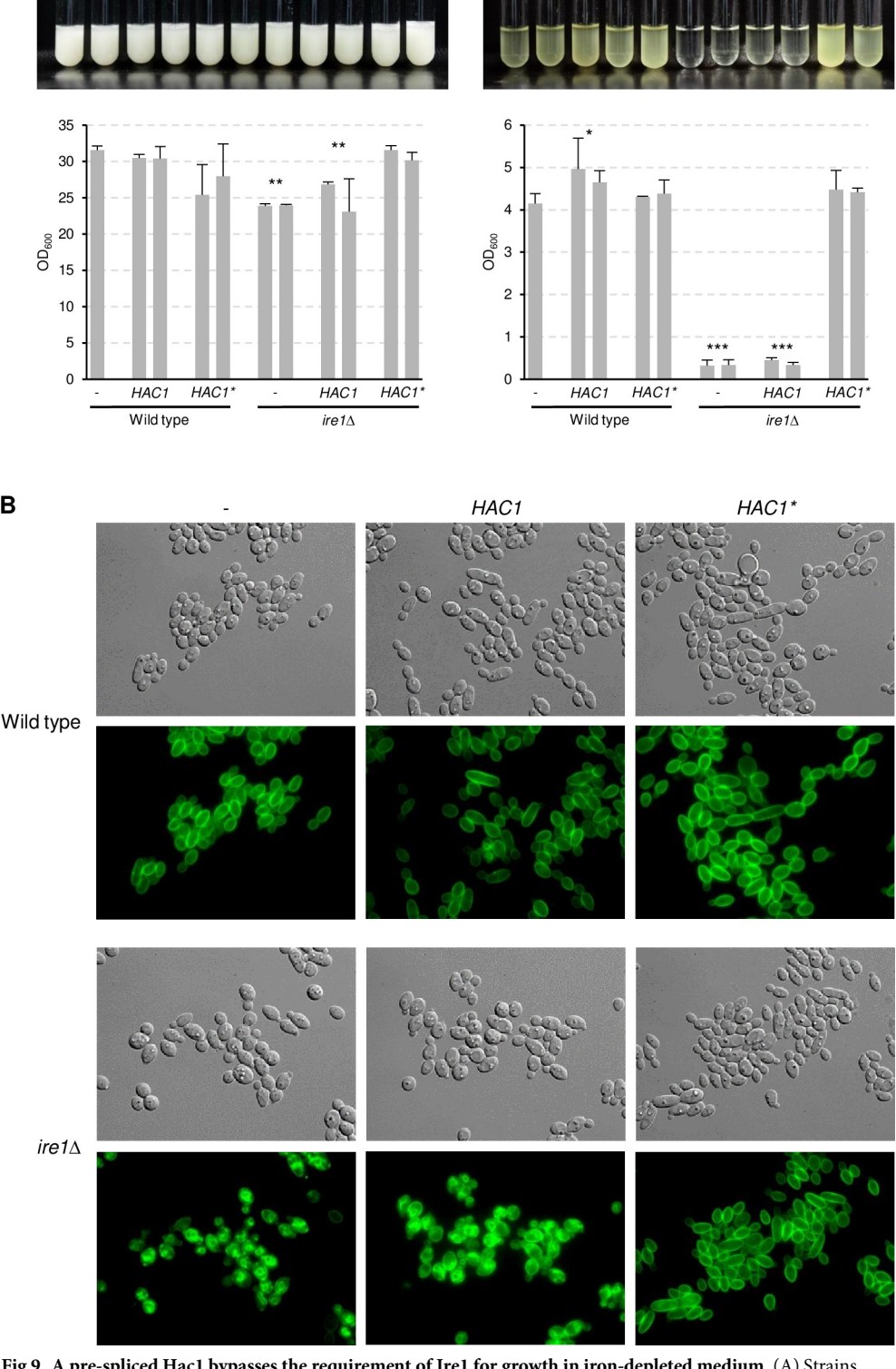

**Fig 9. A pre-spliced Hac1 bypasses the requirement of Ire1 for growth in iron-depleted medium.** (A) Strains expressing wild-type *HAC1* or an intron-less copy (*HAC1\**) from the *ADH1* promoter in wild-type or *ire1Δ* backgrounds were grown in minimal medium with (+ FeCl₃) or without iron (+ BPS). Cultures were photographed after 4 days of growth at 30˚C (top panels) and the optical densities of three independent cultures of each strain

determined (bottom panels showing means and standard deviations). Results for both independently generated series of strains are shown separately; data were combined for statistical analysis. Significant differences from the wild type are indicated by stars (***, p < 0.001; **, p < 0.01; *, p < 0.05. (B) The same strains carrying the *GFP*-tagged *FTR1* allele were grown for 5 h in minimal medium without iron at 30˚C and imaged by DIC and fluorescence microscopy. The two independently generated series of strains behaved identically and only one of them is shown.

concordance with previously reported observations [28]. However, their sensitivity to DTT was less pronounced than that of cells lacking Ire1 and, unlike the *ire1*Δ mutants, the *hac1*Δ mutants were not hypersensitive to caffeine. Furthermore, the *hac1*Δ mutants did not display the temperature sensitivity of *ire1*Δ mutants (S1 Fig), a phenotype that has also been observed in *Aspergillus fumigatus* and *Cryptococcus neoformans* mutants with a defective UPR, but not in *Saccharomyces cerevisiae* and *Candida glabrata* [29–32]. All mutant phenotypes were reverted after reintegration of a single *IRE1* or *HAC1* gene copy into the respective mutants (S1 Fig). These results demonstrate that Ire1 has Hac1-independent functions in the adaptation of *C. albicans* to iron limitation and other stress conditions.

## Hac1 contributes to correct Ftr1 localization under conditions of ER stress

The results described above indicated that Ire1 has a basal function in maintaining optimal functionality of the secretory pathway that does not require Hac1 activation in the absence of ER stress. The fact that constitutive Hac1 activation bypassed the requirement of Ire1 for proper Ftr1 localization suggested that Ire1 target proteins that ensure Ftr1 transport under nonstressed conditions are among those that are upregulated by Hac1 in response to ER stress to re-establish ER homeostasis. We hypothesized that under conditions of ER stress, an enhanced activity of these proteins might be necessary to ensure Ftr1 transport and require their upregulation by Hac1. We therefore observed Ftr1 localization in wild-type and *hac1*Δ cells treated with DTT (S2 Fig). Under the experimental conditions used, some Ftr1 was detected in the vacuole in addition to the cell membrane in both wild-type and *hac1*Δ cells treated with DTT, but Ftr1 was also partially retained in the ER in the *hac1*Δ mutants, in line with the established role of Hac1 in maintaining functionality of the secretory pathway under conditions of ER stress.

## A pre-spliced *HAC1* restores virulence in *ire1*Δ mutants

The failure of *ire1*Δ mutants to adapt to iron starvation, together with their hypersensitivity to cell wall/membrane stress and elevated temperatures, suggested that their ability to cause a systemic infection would be strongly attenuated or abolished, as for mutants of other pathogenic fungi lacking this kinase [33]. However, since expression of a pre-spliced *HAC1*, which bypassed the requirement for Ire1 to produce a functional Hac1 transcription factor, reverted growth defects of *ire1*Δ mutants, we wondered if a constitutively active Hac1 would also restore virulence. Furthermore, since the *hac1*Δ mutants did not exhibit the growth defects of the *ire1*Δ mutants under iron-limiting conditions and at elevated temperatures and also displayed milder sensitivity to ER stress (Figs 10 and S1), we wanted to directly compare their virulence. To avoid a potential negative effect of constitutive overexpression of the activated *HAC1**, as has been observed in *S. cerevisiae* [31], we first tested if removal of the intron from one of the endogenous *HAC1* alleles would also revert the *in vitro* growth defects of *ire1*Δ mutants. As can be seen in S3 Fig, expression of *HAC1** at the endogenous genomic locus restored growth of *ire1*Δ mutants in iron-depleted medium, at elevated temperatures, and in the presence of inhibitors as efficiently as when *HAC1** was ectopically expressed from the *ADH1* promoter. These strains were therefore used for infection experiments.

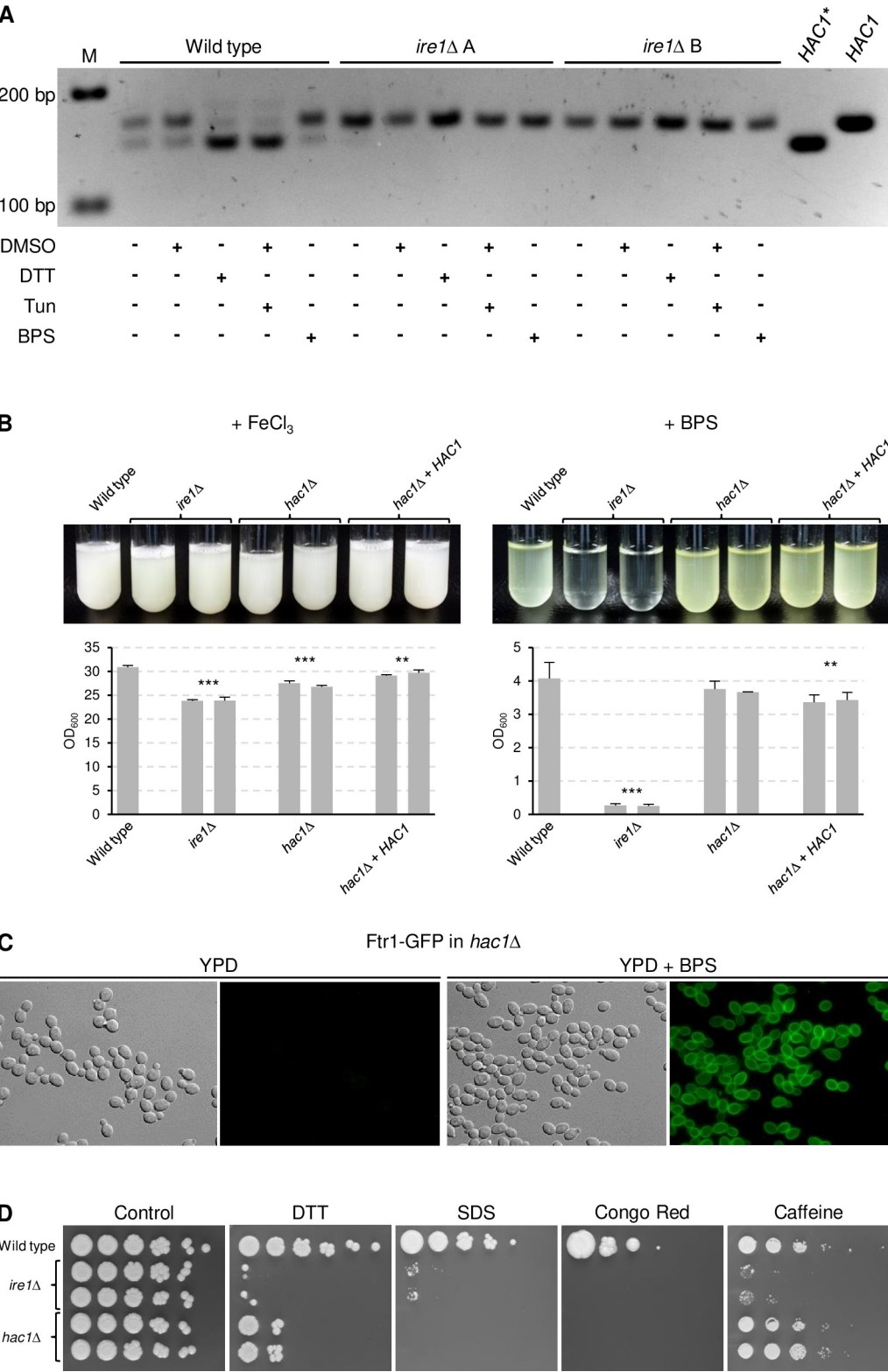

**Fig 10. *HAC1* is not required for growth of *C. albicans* under iron-limiting conditions.** (A) Detection of spliced and unspliced forms of *HAC1* mRNA in the wild-type strain SC5314 and *ire1Δ* mutants. The strains were grown in the absence or presence of 5 mM DTT, 4 μg/ml tunicamycin, or 500 μM BPS as described in Materials and methods. Since tunicamycin is dissolved in DMSO, the cultures were also treated with a corresponding amount of DMSO alone as an

additional control to no treatment. Total RNA was isolated from the cells and the region containing the *HAC1* intron amplified by RT-PCR. The RT-PCR products were analyzed by agarose gel electrophoresis; the expected sizes are 162 bp and 143 bp, respectively, for the unspliced and spliced *HAC1* transcripts. PCR products originating from plasmids containing wild-type *HAC1* or the pre-spliced *HAC1\** are included for comparison in addition to size markers (M). (B) The wild-type strain SC5314, *ire1Δ* mutants, and *hac1Δ* mutants and complemented strains were grown in minimal medium with (+ FeCl₃) or without iron (+ BPS). Cultures were photographed after 4 days of growth at 30˚C (top panels) and the optical densities of three independent cultures of each strain determined (bottom panels showing means and standard deviations). Results for both independently generated series of strains are shown separately; data were combined for statistical analysis. Significant differences from the wild type are indicated by stars (\*\*\*, p < 0.001; \*\*, p < 0.01). (C) Overnight cultures of *hac1Δ* mutants containing a *GFP*-tagged *FTR1* allele were inoculated in fresh YPD medium without or with BPS and grown for 5 h at 30˚C. Cells were imaged by DIC and fluorescence microscopy. (D) Comparison of the sensitivities of *ire1Δ* and *hac1Δ* mutants to ER and cell membrane/wall stress. YPD overnight cultures of the strains were serially 10-fold diluted, spotted on YPD plates without (control) or with 20 mM DTT, 0.04% SDS, 50 μg/ml Congo Red, or 15 mM caffeine and incubated for 4 days at 30˚C. Both independently generated series of mutants are shown in (A), (B), and (D). The two independently generated *hac1Δ* mutants containing Ftr1-GFP behaved identically and only one of them is shown in (C).

As expected, the *ire1Δ* mutants were avirulent in the mouse model of systemic candidiasis, with all mice surviving until the end of the experiment (Fig 11A, left) without developing clinical symptoms, and no viable fungal cells were recovered from the kidneys of infected mice (Fig 11B). Virulence was restored to wild-type levels upon reintroduction of a functional *IRE1* copy into the *ire1Δ* mutants. Importantly, expression of the pre-spliced *HAC1\** also largely restored virulence of the *ire1Δ* mutants; the prolonged survival of mice infected with these strains compared to mice infected with the wild type did not reach statistical significance (Fig 11A, left), and kidney fungal burdens in the two groups of mice were also similar (Fig 11B). Therefore, the reversion of the *in vitro* growth defects of the *ire1Δ* mutants by the pre-spliced *HAC1\** was also reflected by a recovery of virulence in a mammalian host. That virulence was not fully restored is conceivable if the stress conditions encountered during an infection require complete conversion of the *HAC1* mRNA into its spliced form, at least in some environments. This is not achievable in the *ire1Δ* mutants in which one of the two *HAC1* alleles is pre-spliced, but the mRNA produced from the other allele cannot be spliced due to the absence of Ire1. Therefore, these cells might have lower levels of active Hac1 than a wild-type strain.

Similar to the *ire1Δ* mutants, the *hac1Δ* mutants were also avirulent in this infection model, and their virulence defect was reverted in the complemented strains (Fig 11A, right, and Fig 11B). This result indicates that, despite their wild-type growth under iron-limiting conditions and elevated temperatures observed in the *in vitro* experiments, the sensitivity of *hac1Δ* mutants to other stressful conditions rendered them unable to cause a systemic infection.

## Discussion

The initial goal of this study was to gain more insight into how the activity of Sef1, a key transcriptional activator of iron uptake genes that is essential for the adaptation of *C. albicans* to iron limitation, is regulated. When testing if an artificially activated form of Sef1 would bypass the reported requirement of the protein kinase Ssn3 for Sef1 function, we unexpectedly observed that Ssn3 was dispensable for growth under the iron-limiting conditions used in our experiments, in which Sef1 is crucial. Since Sef1 is phosphorylated in response to iron limitation, we generated a comprehensive protein kinase deletion mutant library of the *C. albicans* wild-type reference strain SC5314 to systematically test if other protein kinases are involved in the regulation of Sef1 activity and adaptation to iron limitation. Screening of the mutant library identified three protein kinases that were required for growth in iron-depleted media, but not in iron-replete conditions. However, none of these three protein kinases had an apparent role in regulating Sef1 activity. Vps15 is part of a vacuolar protein sorting complex and

## A

### Survival of mice after systemic infection

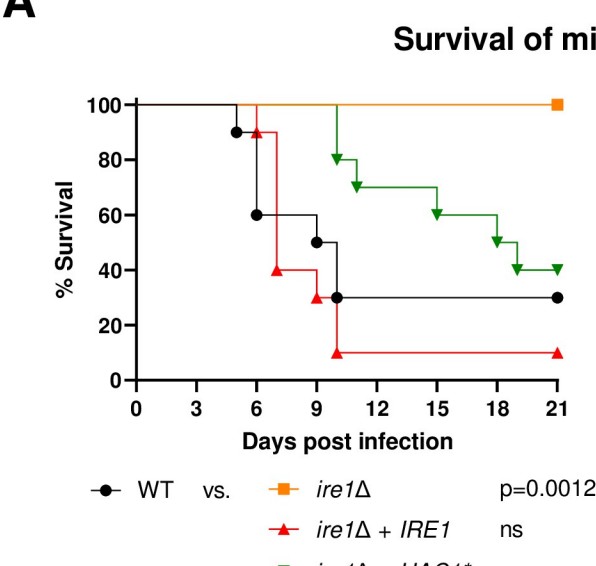

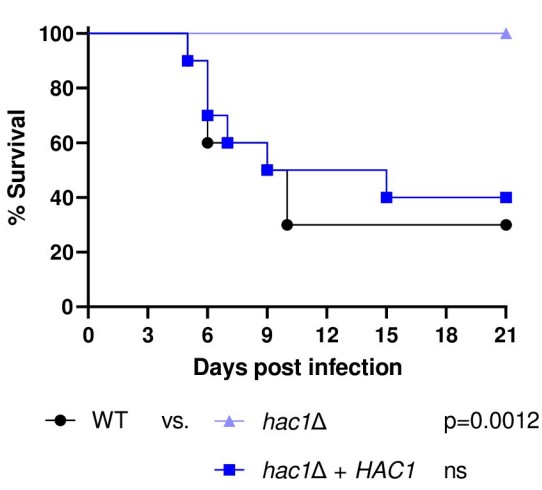

## B

### Fungal burden

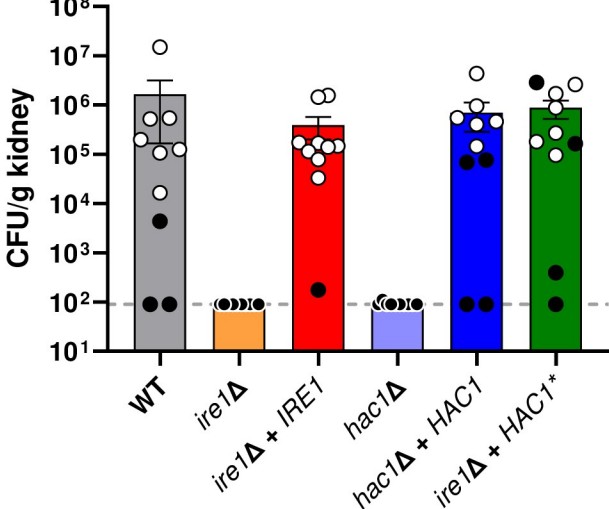

| WT vs. | *ire1Δ* | p=0.0102 |
| | *ire1Δ + IRE1* | ns |
| | *hac1Δ* | p=0.0176 |
| | *hac1Δ + HAC1* | ns |
| | *ire1Δ + HAC1** | ns |

**Fig 11. A pre-spliced *HAC1* restores virulence in *ire1Δ* mutants.** (A) Kaplan-Meyer curves of mouse survival after infection with the wild-type strain SC5314, *ire1Δ* mutants and complemented strains (*ire1Δ + IRE1*), *hac1Δ* mutants and complemented strains (*hac1Δ + HAC1*), and *ire1Δ* mutants containing an intron-less copy of *HAC1* (*ire1Δ + HAC1**). Pooled data from two independent experiments in which the wild type and the complete A or B series of mutants were tested in parallel (five mice per strain and experiment) are shown. Data are presented in two separate graphs for ease of comparison, with the data for the wild type included in both graphs. Pairwise comparison was performed using the Log-rank test; ns, not significantly different from the wild type. (B) Fungal burden in the kidneys of mice shown in (A), determined by plating of organ homogenates and calculation of CFU/g tissue. The horizontal line indicates the detection limit (90 CFU/g); for statistical analysis, samples from which no colonies were obtained were set to the detection limit. White circles represent data from animals reaching humane endpoints, black circles indicate mice sacrificed 21 days after infection. Bars represent mean and SEM. Data were analyzed by 1-way ANOVA followed by Kruskal-Wallis and Dunn's multiple comparisons test to compare each strain to the wild type.

required for normal vacuolar morphology and retrograde protein trafficking [34]. *C. albicans vps15Δ* mutants are hypersensitive to a variety of stress conditions [34], and the *vps15Δ* mutants generated in our present study from the wild-type strain SC5314 also exhibited

reduced growth in control media. The inability of the *vps15Δ* mutants to grow in iron-depleted media may therefore reflect a general requirement for Vps15 to thrive under harsh conditions rather than a specific role in adaptation to iron limitation. Mec1 is a cell cycle checkpoint kinase required for genome stability, and *C. albicans mec1Δ* mutants are hypersensitive to DNA-damaging chemicals [35]. Compared to the *ire1Δ* mutants, the *mec1Δ* mutants had a much milder growth defect in iron-depleted media (see Fig 3), the basis of which remains unknown. In contrast, for Ire1 we uncovered why this protein kinase is essential for the ability of *C. albicans* to adapt to iron limitation: Ire1 is required for proper localization of the high-affinity iron permease Ftr1 to the cell membrane.

Ire1 is best known for its conserved function in the unfolded protein response of eukaryotic cells [25], and it is required for cell wall biogenesis and morphogenesis in *C. albicans* [19]. Ire1 is an ER-resident type I transmembrane protein that consists of an N-terminal lumenal domain and a cytosolic part that contains both a serine/threonine kinase domain and a C-terminal endoribonuclease (RNase) domain. Upon interaction of its lumenal domain with unfolded proteins during ER stress, Ire1 oligomerizes and *trans*-autophosphorylates to activate its RNase function, resulting in *HAC1* mRNA splicing. The mature Hac1 transcription factor then activates its target genes, including genes encoding ER chaperones that help in protein folding, to restore ER homeostasis [25].

A plausible explanation for the inability of the *ire1Δ* mutants to properly transport Ftr1 to the cell membrane would have been that iron starvation causes ER stress, to which cells lacking Ire1 cannot respond. However, iron depletion did not provoke the UPR, since it did not cause increased *HAC1* mRNA splicing. In line with this, a recent study also found that iron limitation *per se* did not cause ER stress in *S. cerevisiae*, although these authors used milder conditions of iron limitation [36]. Therefore, Ire1-dependent basal *HAC1* mRNA splicing seemed to ensure optimal ER functionality and Ftr1 transport in the absence of ER stress. In *A. fumigatus*, *hacA* mRNA splicing also occurs under non-stressed conditions, indicating that the UPR buffers minor fluctuations in ER stress that arise during normal growth [30]. Similarly, in *C. neoformans* basal levels of *HXL1* (the functional equivalent of *HAC1*) mRNA splicing occur under noninducing conditions, although the encoded protein could only be detected under conditions of ER stress [29]. Surprisingly, however, we found that Hac1 was not required for normal Ftr1 localization and growth in iron-depleted media. These results demonstrate that Ire1 has a Hac1-independent role in the maintenance of ER functionality. Indeed, in *A. fumigatus*, IreA has both HacA-dependent and HacA-independent functions [30], and Ire1 in *C. neoformans* also has functions outside of the canonical UPR [29]. In *C. glabrata*, *HAC1* mRNA splicing does not occur and Ire1 activates the UPR in a Hac1-independent manner, in cooperation with other signaling pathways [31]. Proteins whose activity is directly controlled by Ire1 remain to be identified (currently, Ire1 itself is the only validated phosphorylation target of the kinase [37]). Our finding that a constitutively active Hac1 bypasses the defects of *ire1Δ* mutants suggests that the relevant Ire1 target proteins are among those that are upregulated during ER stress in a Hac1-dependent fashion to restore functionality of the secretory pathway. The dependence of Ftr1 on a functional Ire1 kinase for correct localization even in the absence of ER stress may be related to the fact that Ftr1 associates with a ferroxidase already early in the secretory pathway and the two proteins are transported together to the plasma membrane [8,38,39]. This associated transport may be more sensitive to a compromised ER function than the trafficking of other transporters, such as Mep2.

A role for Ire1 in growth under iron-limiting conditions has been observed previously in *A. fumigatus* [30]. Mutants with a defect in the UPR (both *ΔireA* and *ΔhacA* mutants) also exhibited a growth defect under iron-limiting conditions, a phenotype that was explained by reduced expression of genes involved in siderophore-mediated iron acquisition and reductive

iron assimilation. In contrast, Ire1 was not required for growth in iron-depleted medium in *C. glabrata* [31]. Very recently, Sircaik *et al*., in an extensive functional analysis of *C. albicans* Ire1, reported that an *ire1* knock-down mutant had a growth defect on iron-depleted medium, which the authors associated with reduced expression of iron homeostasis genes in the mutant compared to a wild-type strain during growth in iron-replete rich medium [40]. However, we found that genes that are known to be induced by Sef1 in response to iron limitation [3] were efficiently upregulated in *ire1Δ* mutants generated from the wild-type strain SC5314. We note that the mRNA levels of these genes were slightly lower in the *ire1Δ* mutants than in the wild type after 5 h of growth in iron-depleted medium (see Fig 5A), but this is likely due to the growth defect of the *ire1Δ* mutants in this medium, which results in lower amounts of newly produced mRNAs compared to those of constitutively expressed genes such as the *ACT1* control. We conclude that the essential role of Ire1 in the adaptation of *C. albicans* to iron limitation is not caused by a defect in the expression of iron uptake genes but by its requirement for transport of the crucial high-affinity iron transporter Ftr1 to the cell membrane.

The sensitivity of *ire1Δ* mutants to many stress conditions predicted that their virulence in a mammalian host would be strongly attenuated. This was indeed the case and was also reported in the recent study by Sircaik *et al.* [40]. Our finding that *hac1Δ* mutants were similarly avirulent in the mouse model of systemic candidiasis argues that a failure to properly respond to ER stress renders *C. albicans* unable to cause a disseminated infection. However, cells lacking Ire1 have additional deficiencies that are not exhibited by *hac1Δ* mutants and affect traits that are important for virulence, such as growth at the mammalian body temperature and under iron-limiting conditions. The inability of *ire1Δ* mutants to ensure functionality of the high-affinity iron permease Ftr1, a well-established virulence factor that is essential for growth in iron-restricted host environments [5], demonstrates that Ire1 also has a Hac1-independent role in virulence in *C. albicans*.

## Materials and methods

### Ethics statement

Mice were cared for in accordance with the European Convention for the Protection of Vertebrate Animals Used for Experimental and Other Scientific Purposes. The animal experiments were performed in accordance with European and German regulations. Protocols were approved by the Thuringian authority and ethics committee (Thüringer Landesamt für Verbraucherschutz, permit number HKI-21-006).

### Strains and growth conditions

The *C. albicans* strains used in this study are listed in S1 and S2 Tables. All strains were stored as frozen stocks with 17.2% glycerol at -80˚C. Strains were routinely grown in YPD liquid medium (1% yeast extract, 2% peptone, 2% glucose) at 30˚C in a shaking incubator. To prepare YPD agar plates, 1.5% agar was added before autoclaving. For selection of nourseothricin-resistant transformants, 200 μg/ml nourseothricin (Werner Bioagents, Jena, Germany) was added to YPD agar plates. To obtain nourseothricin-sensitive derivatives in which the *SAT1* flipper cassette was excised by FLP-mediated recombination, transformants were grown overnight in YCB-BSA-YE medium (2.34% yeast carbon base, 0.4% bovine serum albumin, 0.2% yeast extract, pH 4.0) without selective pressure to induce the *SAP2* promoter controlling *caFLP* expression. Appropriate dilutions were plated on YPD agar plates and grown for 2 days at 30˚C. Individual colonies were picked and streaked on YPD plates as well as on YPD plates with 100 μg/ml nourseothricin to confirm nourseothricin-sensitivity.

## Plasmid constructions

For gene deletions, ca. 0.5 kb of the upstream and downstream regions of the target genes were amplified from genomic DNA of strain SC5314 (oligonucleotide primers and their usage are listed in S3 Table), digested at the introduced SacI/SacII and XhoI/ApaI sites, respectively, and cloned on both sides of the *SAT1* flipper cassette in plasmid pSFS5 [41]. For reintroduction of functional gene copies into deletion mutants, the complete coding regions and flanking sequences were amplified as SacI-SacII fragments (in some cases in two parts) and inserted in place of the upstream flanking region in the corresponding deletion constructs. To express *SEF1* from the *ADH1* promoter, the *SEF1* coding sequence was amplified by PCR and substituted for the *UPC2* coding sequence in the previously described plasmid pUPC2E2 [15], resulting in pSEF1E1. To generate a 3xHA-tagged Sef1, a part of the *SEF1* coding region was amplified with primers that introduced a BamHI site (encoding a Gly-Ser linker) instead of the stop codon. The PCR product was ligated together with a fragment from pCEK1H1 [42], which contains the 3xHA sequence followed by a stop codon and the *ACT1* transcription termination sequence, and inserted instead of the *SEF1* upstream sequence in the *SEF1* deletion cassette to produce pSEF1H2. For GFP-tagging of Ftr1, a part of the *FTR1* coding region was amplified with primers that introduced a KasI site (encoding a Gly-Ala linker) instead of the stop codon. The *FTR1* downstream sequence was also amplified and the two PCR products were inserted instead of *YOR1* sequences in the previously described plasmid pYOR1G1 [43] to generate pFTR1G2. To express *HAC1* from the *ADH1* promoter, the *HAC1* coding sequence was amplified by PCR and substituted for the *UPC2* coding sequence in pUPC2E2, producing pHAC1E1. An intronless *HAC1* copy was generated by fusion PCR with primers that amplified the regions upstream and downstream of the intron and cloned in an analogous fashion to yield pHAC1exE1. The intronless *HAC1* was also amplified together with flanking sequences and inserted instead of the *HAC1* upstream sequence in the *HAC1* deletion cassette to obtain pHAC1ex1K2, which was used to replace one of the endogenous *HAC1* alleles by a pre-spliced copy.

## *C. albicans* strain construction

*C. albicans* strains were transformed by electroporation [22] with gel-purified inserts from the plasmids described above. Plasmid pSEF1-GAD1 containing the artificially activated *SEF1* has been described previously [15]. For introducing a *GFP*-tagged *MEP2*, the insert from the previously described pMEP2G7 [44] was used. The correct genomic integration of all constructs and excision of the *SAT1* flipper cassette were confirmed by Southern hybridization using the flanking sequences as probes.

## Isolation of genomic DNA and Southern hybridization

Genomic DNA from *C. albicans* strains was isolated as described previously [22]. The DNA was digested with appropriate restriction enzymes, separated on a 1% agarose gel, transferred by vacuum blotting onto a nylon membrane, and fixed by UV crosslinking. Southern hybridization with enhanced chemiluminescence-labeled probes was performed with the Amersham ECL Direct Nucleic Acid Labelling and Detection System (GE Healthcare UK Limited, Little Chalfont Buckinghamshire, UK) according to the instructions of the manufacturer.

## Growth assays

To assess growth and riboflavin secretion in iron-depleted medium, YPD overnight cultures of the strains were diluted to an optical density at 600 nm ($OD_{600}$) of 2.0 in water. Thirty

microliters of the cell suspensions was transferred to 3 ml of test and control media. We used a modified SD minimal medium (0.17% yeast nitrogen base [YNB] w/o ammonium sulfate, w/o copper sulfate, w/o ferric chloride [MP Biomedicals, Illkirch, France], 0.079% complete supplement medium [CSM, MP Biomedicals], 2% glucose, 0.5% ammonium sulfate, 1 μM copper sulfate) to which either 200 μg/L ferric chloride or 100 μM BPS was added. After 4 days of growth at 30˚C in a shaking incubator, the cultures were photographed and the $OD_{600}$ determined. The sensitivity of strains to iron limitation and various other types of stress was also tested by dilution spot assays on solid media. For this, YPD overnight cultures were diluted to an $OD_{600}$ of 2.0 and tenfold dilutions from $10^0$ to $10^{-5}$ prepared in a 96-well microtiter plate. Ca. 5 μl of the cell suspensions was transferred with a replicator onto SD agar plates without or with 3 mM $H_2O_2$ or YPD plates without or with 300 μM BPS, 50 μg/ml Congo Red, 15 mM caffeine, 0.04% SDS, or 10 mM DTT. The plates were incubated for 4 days at 30˚C and photographed.

## Northern hybridization

Overnight cultures of the strains were diluted to an $OD_{600}$ of 0.4 in fresh YPD medium without or with 500 μM BPS and grown for 5 h at 30˚C. Total RNA was extracted using a Quick-RNA Fungal/Bacterial Miniprep Kit (Zymo Research, Irvine, CA) following the manufacturer's instructions. RNA samples were separated on a 1.2% agarose gel, transferred by capillary blotting onto a nylon membrane, and fixed by UV crosslinking. The blots were hybridized with PCR-amplified, digoxigenin-labeled probes for *ACT1* (positions +1103 to +1591 in the coding sequence), *CSA1* (positions +2040 to +2477), *FET31* (positions +666 to +921), *FRP1* (positions +271 to +738), *FTR1* (positions +53 to + 259), and *RIB1* (positions +427 to +793). Bound probe was detected with a peroxidase-labeled anti-digoxigenin AP-conjugate (Roche, Basel, Switzerland). Signals were generated using CSPD (Roche, Basel, Switzerland) as substrate and captured with the ImageQuant LAS 4000 imaging system (GE Healthcare).

## Western blotting

Overnight cultures of the strains were diluted $10^{-2}$ in 50 ml fresh YPD medium without or with 500 μM BPS and grown for 5 h at 30˚C. Cells were collected by centrifugation, washed in 1 ml ice-cold water, and resuspended in 300 μl breaking buffer (100 mM triethylammonium bicarbonate buffer [TEAB], 150 mM NaCl, 1% SDS, cOmplete EDTA-free Protease Inhibitor Cocktail and PhosStop Phosphatase Inhibitor Cocktail [Roche Diagnostics GmbH, Mannheim, Germany]). An equal volume of 0.5 mm acid-washed glass beads was added to each tube. Cells were mechanically disrupted on a FastPrep-24 cell-homogenizer (MP Biomedicals, Santa Ana, USA) with three 40 s pulses, with 5 min on ice between each pulse. Samples were centrifuged at 21,000 x *g* for 15 min at 4˚C, the supernatant was collected, and the protein concentration was quantified using the Bradford protein assay. Equal amounts of protein of each sample were mixed with one volume of 2x Laemmli buffer, heated for 5 min at 95˚C, and separated on an SDS-9% polyacrylamide gel. Separated proteins were transferred onto a nitrocellulose membrane with a mini-Protean System (Bio-Rad, Munich, Germany) and stained with Ponceau S to control for equal loading. For the detection of HA-tagged Sef1, membranes were blocked in TBST with 5% milk and incubated overnight with rat monoclonal anti-HA-Peroxidase antibody (Roche Diagnostics GmbH, Mannheim, Germany). GFP-tagged Ftr1 was detected using an anti-GFP antibody (Abcam goat polyclonal antibody to GFP (HRP) ab 6663). Membranes were washed in Tris-buffered saline with Tween 20 (TBST) and signals detected with the ECL chemiluminescence detection system (GE Healthcare Bio-Sciences GmbH, Munich, Germany).

## Fluorescence microscopy

Overnight cultures of strains with the *GFP*-tagged *FTR1* were diluted $10^{-2}$ in 50 ml fresh YPD medium without or with 500 μM BPS and grown for 5 h at 30˚C. For ER staining, 500 μl of the cultures was centrifuged after 4 h and the cells were resuspended in ER detection reagent (Abcam Cytopainter ER staining kit) and incubated for further 30 min at 30˚C. The cells were washed once with Assay buffer and imaged with a Leica DMI6000 fluorescence microscope using appropriate filters for red and green fluorescence detection. For vacuole staining, 1 ml of each culture was centrifuged after 1 h and the cells were resuspended in 100 μl YPD with 20 μM FM 4–64 and incubated for 40 min at 30˚C. The cells were washed with 1 ml YPD, resuspended in 5 ml YPD or YPD with 500 μM BPS, and incubated for 4 h at 30˚C before being observed by fluorescence microscopy. Strains with the *GFP*-tagged *MEP2* were grown for 5 h at 30˚C in standard SD medium (0.67% YNB with ammonium sulfate, 2% glucose) or in SLAD medium (0.17% YNB w/o ammonium sulfate, 50 μM ammonium sulfate, 2% glucose). To localize the Mep2-GFP and Ftr1-GFP proteins under conditions of both iron depletion and nitrogen starvation, the corresponding strains were also grown in a modified SLAD medium (0.17% YNB w/o ammonium sulfate, w/o copper sulfate, w/o ferric chloride, 2% glucose, 50 μM ammonium sulfate, 1 μM copper sulfate, 100 μM BPS).

## Detection of spliced and unspliced *HAC1* transcripts by RT-PCR

Overnight cultures of the strains were diluted to an $OD_{600}$ of 0.4 in fresh YPD medium and incubated for 3 h at 30˚C, at which point 500 μM BPS, 5 mM DTT, 4 μg/ml tunicamycin, or DMSO alone was added, and the cultures incubated for an additional hour. Total RNA was extracted, treated with DNase, and cDNA synthesized. RT-PCR was performed using primers HAC1.10 (5'-ATCATCAACCTCCCCTTCCT-3', positions +945 to +964 in *HAC1*) and HAC1.11 (5'-TCAACATCATCTCCTAAAATCGAA-3', positions +1107 to +1084 in *HAC1*), which bind upstream and downstream of the *HAC1* intron. PCR fragments were separated in a 4% agarose gel prepared with 0.5x TBE buffer.

## Mouse model of systemic candidiasis

Eight- to ten-week-old female specific-pathogen-free BALB/c mice (16 to 18 g), purchased from Charles River (Germany), were housed in groups of five in individually ventilated cages at 22 ± 1˚C, 55 ± 10% relative humidity, 12 h/12 h dark/light cycle, with free access to food and water, bedding material, and autoclavable mouse houses as environmental enrichment. Body weight and body surface temperature were recorded daily. Mice were infected with 1 x $10^4$ CFU/g body weight in 100 μl Dulbecco's Phosphate Buffered Saline (DPBS, Gibco) via the lateral tail vein at day 0. To obtain the infection inoculum, fresh *C. albicans* colonies grown on YPD agar plates were inoculated into liquid YPD and grown to late exponential phase (14 to 16 h) at 30˚C with horizontal shaking at 180 rpm. Cells were washed twice with sterile PBS and resuspended in DPBS, counted using a hemocytometer, and adjusted to the desired concentrations in DPBS. Infectious doses were confirmed by serial dilutions and plating on YPD agar plates.

After infection, the health status of the mice was checked at least twice a day and an additive clinical score was determined to evaluate disease severity. The following parameters were included: body weight (increase or decrease ≤ 5%, 0; decrease 5% - 20% from the initial weight, 1; decrease ≥ 20% of the initial weight, 2), fur (normal, 0; slightly ruffled, 1; ruffled, 2), lethargy (absent, 0; mild, 1; moderate, 2; severe, 3), posture (normal, 0; high-legged gait <u>or</u> hunched back while sitting, 1; high-legged gait <u>and</u> hunched back while sitting and moving, 2), body temperature (normal, 0; moderately increased, 1; increased, 2; hypothermia, 3). The

maximum possible score was 12. Mice were euthanized with an overdose of ketamine (100 μl of 100 mg/ml) and xylazine (25 μl of 20 mg/ml) applied intraperitoneally when they reached the humane endpoints defined as (i) severe lethargy, (ii) hypothermia, or (iii) a cumulative clinical score of > 5. Mice without severe illness were sacrificed on day 21. Following euthanasia, kidneys were removed during autopsy, weighed, and homogenized in PBS using an Ultra-Turrax (Ika). Homogenates were serially diluted and plated onto YPD plates containing 80 μg/ml chloramphenicol (Roth) for enumeration of CFU. Two independent experiments were performed (one with the wild type and the A series of all mutants tested in parallel, and one with the wild type and the B series of all mutants), with five mice per strain and experiment. Data from the two experiments were combined for the statistical analyses.

## Statistical analyses

*In vitro* growth differences between strains were analyzed by one-way ANOVA followed by Dunnett's test for comparing several treatments with a control (95% family-wise confidence level). For the analysis of the mouse infection experiments, data were plotted and analyzed using GraphPad Prism version 9.2 (GraphPad Software, San Diego, USA). Survival data were plotted as Kaplan-Meyer curves and analyzed by Log-rank test. Fungal burden was analyzed by 1-way ANOVA followed by the Kruskal-Wallis and Dunn's multiple comparisons test for multiple comparison (assuming nonparametric distribution based on analysis by tests for normal distribution provided in GraphPad Prism).

## Supporting information

**S1 Fig. Sensitivity of *ire1Δ* and *hac1Δ* mutants and complemented strains to ER and cell membrane/wall stress and elevated temperatures.**
(PDF)

**S2 Fig. Hac1 contributes to correct Ftr1 localization under conditions of both iron limitation and ER stress.**
(PDF)

**S3 Fig. Removal of the intron in one of the endogenous *HAC1* alleles is sufficient to rescue growth defects of *ire1Δ* mutants.**
(PDF)

**S1 Table. *Candida albicans* protein kinase deletion mutant library.**
(XLSX)

**S2 Table. *Candida albicans* strains used in this study.**
(XLSX)

**S3 Table. Oligonucleotide primers used for plasmid constructions.**
(XLSX)

## Author Contributions

**Conceptualization:** Bernardo Ramírez-Zavala, Joachim Morschhäuser.

**Data curation:** Bernardo Ramírez-Zavala, Christine Dunker, Ilse D. Jacobsen, Joachim Morschhäuser.

**Formal analysis:** Bernardo Ramírez-Zavala, Christine Dunker, Ilse D. Jacobsen, Joachim Morschhäuser.

**Funding acquisition:** Ilse D. Jacobsen, Joachim Morschhäuser.

**Investigation:** Bernardo Ramírez-Zavala, Ines Krüger, Christine Dunker, Ilse D. Jacobsen, Joachim Morschhäuser.

**Methodology:** Bernardo Ramírez-Zavala, Christine Dunker, Ilse D. Jacobsen, Joachim Morschhäuser.

**Project administration:** Joachim Morschhäuser.

**Resources:** Joachim Morschhäuser.

**Supervision:** Bernardo Ramírez-Zavala, Ilse D. Jacobsen, Joachim Morschhäuser.

**Validation:** Bernardo Ramírez-Zavala, Christine Dunker, Ilse D. Jacobsen, Joachim Morschhäuser.

**Visualization:** Bernardo Ramírez-Zavala, Christine Dunker, Ilse D. Jacobsen, Joachim Morschhäuser.

**Writing – original draft:** Joachim Morschhäuser.

**Writing – review & editing:** Bernardo Ramírez-Zavala, Christine Dunker, Ilse D. Jacobsen, Joachim Morschhäuser.

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
