## [Decision Letter · Decision Letter 0]

9 Dec 2021

Dear Dr. Morschhäuser,

Thank you very much for submitting your manuscript "The protein kinase Ire1 has a Hac1-independent essential role in iron uptake and virulence of Candida albicans" for consideration at PLOS Pathogens. As with all papers reviewed by the journal, your manuscript was reviewed by members of the editorial board and by several independent reviewers. In light of the reviews (below this email), we would like to invite the resubmission of a significantly-revised version that takes into account the reviewers' comments.  

While all 3 reviewers thought that the work was significant, reviewers 1 and 3 had concerns about the conclusion that Ire1 has hac1-independent functions in Fe homeostasis.  Reviewer 3 was also concerned that the conclusion that iron depletion does not lead to the ER membrane stress is not well-supported by the data.

We cannot make any decision about publication until we have seen the revised manuscript and your response to the reviewers' comments. Your revised manuscript is also likely to be sent to reviewers for further evaluation.

Sincerely,

Scott G. Filler, M.D.

Guest Editor

PLOS Pathogens

Alex Andrianopoulos

Section Editor

PLOS Pathogens

Kasturi Haldar

Editor-in-Chief

PLOS Pathogens

orcid.org/0000-0001-5065-158X

Michael Malim

Editor-in-Chief

PLOS Pathogens

orcid.org/0000-0002-7699-2064

Reviewer's Responses to Questions

**Part I - Summary**

Reviewer #1: This is an interesting analysis of the contribution of the Ire1 UPR sensor to the biology and virulence of Candida albicans, focusing on the role of Ire1 in growth under Fe-limiting conditions. Using a protein kinase deletion mutant library, the Ire1 transmembrane kinase was found to be essential for growth in Fe-depleted media, but surprisingly independent of the Sef1 transcription factor involved in the adaptation to low Fe. The results demonstrate that the high-affinity plasma membrane Fe permease Ftr1 is mislocalized to the ER instead of the plasma membrane in the ire1 mutant. This was not a generalized effect on nutrient transporters since Mep2 was unaffected by ire1 deletion. Interestingly, Fe limitation did not enhance ER stress signaling from the Ire1 sensor, and the gene encoding the Hac1 transcription factor that is activated by Ire1 was dispensable for the movement of Ftr1 to the plasma membrane. However, expression of a pre-spliced copy of HAC1 mRNA in the ire1 mutant bypassed the need for Ire1 with respect to proper localization of Ftr1 and growth in low Fe. Finally, the ire1 and hac1 mutants were avirulent, indicating that C. albicans, like many other pathogenic fungi, rely heavily on ER stress responses in the host environment.

This is an interesting body of work and the logical flow of the data presentation made the manuscript a pleasure to read. The major strength is that the study uncovers a novel aspect of UPR function that involves the localization of a high-affinity Fe permease to the plasma membrane, providing an explanation for how this pathway contributes to the adaptive response to low Fe conditions. The major weakness is that the interpretation of the data in the context of Ire1 having hac1-independent functions in this aspect of Fe homeostasis needs some clarification.

Reviewer #2: Ramírez-Zavala et al. generated and employed a deletion mutant library for protein kinase genes in Candida albicans to identify a role for the Ire1 kinase in growth upon iron deprivation. The activity of Ire1 is independent of the iron regulator Sef1 and instead influences localization of iron uptake functions. These activities are independent of the transcription factor Hac1 that is activated by Ire1 although, intriguingly, expression of a pre-spliced version of Hac1 rescued the iron related phenotypes of the ire1 mutant. Importantly, Ire1 and Hac1 are required for systemic candidiasis. The study nicely builds on previous interesting findings on the connection between the Ssn3 kinase and the iron regulator Sef1, and provides novel new information on iron regulation and Ire1 in Candida albicans. The data are clearly presented and the manuscript is well written.

Reviewer #3: The major finding of the study is that the protein kinase Ire1 has a role for C. albicans growth under iron depletion condition and is required for virulence. However, this is recently published (ref. #37. Figure 5 showed growth defect of an ire1 mutant on YPD+BPS and reduced virulence).

This study also concludes that iron limitation does not cause increased endoplasmic reticulum stress because HAC1 intron removal was not observed under iron depletion condition. Furthermore, the hac1 mutant was able to grow under the iron depletion condition. But Ire1 is known to also mediate other ER membrane stresses that are independent of Hac1. Whether and how iron depletion affects ER membrane stress needs further experiments.

Another major finding of this study is that Ire1 is required for the plasma membrane localization of the iron permease Ftr1 under iron depletion condition. This has been used as a read out for Hac1-independent Ire1 activation in this study. This is a new observation, but it is not clear how Ire1 is regulated by iron depletion. Are there other downstream effectors of Ire1 that contribute to growth under low iron conditions?

**Part II – Major Issues: Key Experiments Required for Acceptance**

Reviewer #1: None

Reviewer #2: None noted

Reviewer #3: 1. This study does not have any experiments that can directly link iron depletion to Ire1 activation. Because HAC1 intron is not removed upon iron depletion, this study concluded that iron depletion does not lead to the ER membrane stress. This conclusion is questionable based on literature. Ire1 affects azole susceptibility in C. albicans (ref. 37) and S. cerevisiae (PMCID: PMC5665437). The S. cerevisiae paper showed a cross-talk between the ER stress signaling and iron/heme homeostasis. Based on the S. cerevisiae study, there are several experiments to determine if similar cross-talk exists in C. albicans. For example, does Sef1 translocate to the nucleus upon UPR activation by DTT. Does heme limitation affect Ftr1 localization to the PM? Is sterol biosynthesis required for Ftr1 PM localization?

2. Why can a pre-spliced HAC1 bypass the requirement of Ire1 for Ftr1 PM localization and for growth under iron depleted conditions when Hac1 is not required for either. This data would suggest that the iron uptake is not completely Hac1 independent. Why the pre-spliced HAC1 completely bypassed the ire1 requirement for growth under iron depleted conditions in vitro, but did not fully bypassed the virulence defect of the ire1 in vivo?

3. The ire1 mutant seemed to have reduced transcription levels of FTR1 and FET31 (Fig. 5A). This is consistent with transcriptional profiling of WT and ire1 by Sircaik et al (PMCID: PMC5665437). But the authors concluded no change? Does the ire1 mutant affect Sef1 nuclear localization under iron depletion condition?

4. The introduction provides no background on Ire1. The result starts with several figures that basically show negative data or data not related to Ire1. Most of these could go into supplemental material. The paper (almost half of the content) starts with experiments that do not contribute to our understanding of how iron depletion regulates Ire1. Real data starts from line 224. The manuscript also missed some key papers, such as PMCID: PMC5665437 and PMCID: PMC7319344.

**Part III – Minor Issues: Editorial and Data Presentation Modifications**

Reviewer #1: There was one aspect that I did not fully understand. The ire1 deletion mutant is defective in Ftr localization & growth in low Fe, which could be rescued by expression of spliced hac1, supporting a role for the Hac1 transcription factor in this process. Since the hac1 deletion mutant did not show this defect, the data are interpreted as evidence that Ire1 has hac1-independent functions in controlling this process (though presumably via different mechanisms). It seems to me that the ability of the processed hac1 to complement the Ftr localization defect in deltaire1 does support a role for Hac1 in this process, which is novel and important. However, I am less convinced that the normal Ftr localization observed in deltahac1 (where ire1 is present but unable to trigger the canonical pathway) necessarily indicates that it is a hac1-independent function of ire1 that is responsible for controlling Ftr localization. My interpretation of the data is that something else is clearly able to ensure proper localization of Ftr and restore growth in low Fe when hac1 is deleted, but this does not necessarily mean that Ire1 is the relevant protein. The data do raise the interesting possibility of involvement of hac1-independent functions of Ire1 playing a role, but other possibilities could also account for this result. This could be addressed by softening the Ire1 conclusion or, if I have misinterpreted the data, clarifying the evidence that clearly supports a role for Ire1 outside of the canonical pathway.

A prior study reported the inability to obtain a complete deletion mutant of Ire1 in C. albicans (Blankenship, PlosPathogens 2010), which prompted a recent study to undertake an ire1 knockdown strategy instead of attempting a homozygous deletion (Sircaik et al.,Cell Microbiol. 2021). I would be curious about the authors thoughts on this apparent difference in essentiality. Does this deletion mutant filament?

Reviewer #2: Line 75. The wording is funny here because it sounds like iron forms a functional permease rather than being the transported substrate.

Line 117. It would help the reader if the SEF1-GAD allele (artificially activated, line 489) was described in more detail here. The activated version of Sef1 is used extensively in the following experiments and it is therefore important to describe its properties thoroughly when it is first introduced.

Line 125, Fig. 1A, B. The strains grow quite poorly on the medium with BPS – especially the ssn3 mutant with the two versions of SEF1. Could it be that the poor growth makes it difficult to evaluate riboflavin production for the ssn3 mutant? A similar question arises for the information on lines 180-181 and the conclusion that no riboflavin is produced by the kinase mutants even though they are growing quite poorly or not at all. Also, is there an explanation for the growth impact of SEF1, particularly in the ssn3 mutant background +BPS in Fig. 1B?.

Of course, this study would be stronger if more insights were obtained into the Hac1-independent roles of Ire1. I wonder if the authors might speculate more about potential targets. For example, are there clues from the cited study of Sircaik et al?

I was intrigued that the pre-spliced version of Hac1 did not restore full virulence to the ire1 mutant. The authors indicate that statistical analysis did not support a difference but it seems that the authors could speculate more about this intermediate result.

Reviewer #3: (No Response)

PLOS authors have the option to publish the peer review history of their article (what does this mean?). If published, this will include your full peer review and any attached files.

Reviewer #1: No

Reviewer #2: No

Reviewer #3: No
---

## [Editor Report · Decision Letter 1]

19 Jan 2022

Dear Dr. Morschhäuser,

We are pleased to inform you that your manuscript 'The protein kinase Ire1 has a Hac1-independent essential role in iron uptake and virulence of Candida albicans' has been provisionally accepted for publication in PLOS Pathogens.

Best regards,

Scott G. Filler, M.D.

Guest Editor

PLOS Pathogens

Alex Andrianopoulos

Section Editor

PLOS Pathogens

Kasturi Haldar

Editor-in-Chief

PLOS Pathogens

orcid.org/0000-0001-5065-158X

Michael Malim

Editor-in-Chief

PLOS Pathogens

orcid.org/0000-0002-7699-2064
---

## [Editor Report · Acceptance letter]

28 Jan 2022

Dear Dr. Morschhäuser,

We are delighted to inform you that your manuscript, "The protein kinase Ire1 has a Hac1-independent essential role in iron uptake and virulence of Candida albicans," has been formally accepted for publication in PLOS Pathogens.

Best regards,

Kasturi Haldar

Editor-in-Chief

PLOS Pathogens

orcid.org/0000-0001-5065-158X

Michael Malim

Editor-in-Chief

PLOS Pathogens

orcid.org/0000-0002-7699-2064